# Hippocampal sharp wave-ripples and the associated sequence replay emerge from structured synaptic interactions in a network model of area CA3

András Ecker[1,2]*[†], Bence Bagi[1,2]‡, Eszter Vértes[1,2]§, Orsolya Steinbach-Németh[1,2], Mária R Karlócai[1], Orsolya I Papp[1], István Miklós[3,4], Norbert Hájos[1], Tamás F Freund[1,2], Attila I Gulyás[1], Szabolcs Káli[1,2]*

[1]Institute of Experimental Medicine, Eötvös Loránd Research Network, Budapest, Hungary; [2]Faculty of Information Technology and Bionics, Pázmány Péter Catholic University, Budapest, Hungary; [3]Alfréd Rényi Institute of Mathematics, Eötvös Loránd Research Network, Budapest, Hungary; [4]Institute for Computer Science and Control, Eötvös Loránd Research Network, Budapest, Hungary

*For correspondence:
andras.ecker@epfl.ch (AE);
kali@koki.hu (SK)

Present address: †Blue Brain Project, École polytechnique fédérale de Lausanne (EPFL), Campus Biotech, Geneva, Switzerland; ‡Department of Bioengineering, Imperial College London, London, United Kingdom; §Gatsby Computational Neuroscience Unit, University College London, London, United Kingdom

**Abstract** Hippocampal place cells are activated sequentially as an animal explores its environment. These activity sequences are internally recreated ('replayed'), either in the same or reversed order, during bursts of activity (sharp wave-ripples [SWRs]) that occur in sleep and awake rest. SWR-associated replay is thought to be critical for the creation and maintenance of long-term memory. In order to identify the cellular and network mechanisms of SWRs and replay, we constructed and simulated a data-driven model of area CA3 of the hippocampus. Our results show that the chain-like structure of recurrent excitatory interactions established during learning not only determines the content of replay, but is essential for the generation of the SWRs as well. We find that bidirectional replay requires the interplay of the experimentally confirmed, temporally symmetric plasticity rule, and cellular adaptation. Our model provides a unifying framework for diverse phenomena involving hippocampal plasticity, representations, and dynamics, and suggests that the structured neural codes induced by learning may have greater influence over cortical network states than previously appreciated.

## Editor's evaluation

This manuscript will be of interest to theoretical neuroscientists broadly defined and potentially also to experimentalists investigating the hippocampus. A biologically plausible spiking neural network model of subregion CA3 of the hippocampus is proposed and studied in order to pinpoint the mechanistic sources of sharp wave ripples and replay observed in vivo.

## Introduction

The hippocampal region plays a pivotal role in spatial and episodic memory (*O'Keefe and Nadel, 1978*; *Morris et al., 1982*). The different stages of memory processing (*Marr, 1971*; *Buzsáki, 1989*) are associated with distinct brain states and are characterized by distinct oscillatory patterns of the hippocampal local field potential (LFP) (*Buzsáki et al., 1983*; *Colgin, 2016*). When rodents explore their environment, place cells of the hippocampus are activated in a sequence that corresponds to the order in which the animal visits their preferred spatial locations (place fields) (*O'Keefe and*

*Dostrovsky, 1971*). The same sequences of firing activity can also be identified, on a faster time scale, during individual cycles of the 4–10 Hz theta oscillation that dominates the hippocampal LFP in this state (*O'Keefe and Recce, 1993*; *Dragoi and Buzsáki, 2006*; *Foster and Wilson, 2007*). These compressed sequences are thought to be optimal for learning via activity-dependent synaptic plasticity (*Jensen and Lisman, 2005*; *Foster and Wilson, 2007*).

Other behavioral states such as slow-wave sleep and quiet wakefulness are characterized by the repetitive but irregular occurrence of bursts of activity in the hippocampus, marked by the appearance of sharp waves (*Buzsáki et al., 1983*; *Wilson and McNaughton, 1994*) and associated high-frequency (150–220 Hz) ripple oscillations (*O'Keefe and Nadel, 1978*; *Buzsáki et al., 1992*) in the LFP. Disruption of SWRs was shown to impair long-term memory (*Girardeau et al., 2009*; *Ego-Stengel and Wilson, 2010*; *Jadhav et al., 2012*; *Oliva et al., 2020*). The activity sequences observed during exploration are also recapitulated during SWRs (*Nádasdy et al., 1999*; *Kudrimoti et al., 1999*; *Lee and Wilson, 2002*), in the absence of any apparent external driver, and this 'replay' can happen either in the same or in a reversed order relative to the original behaviorally driven sequence (*Foster and Wilson, 2006*; *Csicsvari et al., 2007*; *Diba and Buzsáki, 2007*; *Davidson et al., 2009*; *Karlsson and Frank, 2009*; *Gupta et al., 2010*). More specifically, awake replay is predominantly in the 'forward' direction near choice points during navigation (*Diba and Buzsáki, 2007*; *Pfeiffer and Foster, 2013*), while it is mainly 'backward' when the animal encounters a reward (*Diba and Buzsáki, 2007*). Consequently, while sleep replay was suggested to be involved in memory consolidation, forward and reverse replay in awake animals may contribute to memory recall and reward-based learning, respectively (*Carr et al., 2011*; *Foster, 2017*; *Pfeiffer, 2017*; *Olafsdottir et al., 2018*). Finally, behavioral tasks with a high memory demand led to an increase in the duration of SWRs, while artificial prolongation of SWRs improved memory (*Fernández-Ruiz et al., 2019*).

For several decades, area CA3 of the hippocampus has been proposed to be a critical site for hippocampal memory operations, mainly due to the presence of an extensive set of modifiable excitatory recurrent connections that could support the storage and retrieval of activity patterns and thus implement long-term memory (*Marr, 1971*; *McNaughton and Morris, 1987*; *Levy, 1996*; *Rolls, 1996*; *Káli and Dayan, 2000*). Area CA3 was also shown to be strongly involved in the generation of SWRs and the associated forward and reverse replay (*Buzsáki, 2015*; *Davoudi and Foster, 2019*).

Several modeling studies have addressed either theta oscillogenesis, learning, sequence replay or ripples; however, a unifying model of how the generation of SWRs and the associated neuronal activity are shaped by previous experience is currently lacking. In the present study, we built a minimal, yet data-driven model of the CA3 network, which, after learning via a symmetric spike-timing-dependent synaptic plasticity rule (*Mishra et al., 2016*) during simulated exploration, featured both forward and backward replay during autonomously generated sharp waves, as well as ripple oscillations generated in the recurrently connected network of perisomatic interneurons. After validating the model against several in vivo and in vitro results (*Buzsáki et al., 1992*; *Hájos et al., 2013*; *English et al., 2014*; *Schlingloff et al., 2014*; *Stark et al., 2014*; *Pfeiffer and Foster, 2015*; *Gan et al., 2017*), we took advantage of its in silico nature that made feasible a variety of selective manipulations of cellular and network characteristics. Analyzing the changes in the model's behavior after these perturbations allowed us to establish the link between learning from theta sequences and the emergence of SWRs during 'offline' states, to provide a possible explanation for forward and backward replays, and to disentangle the mechanisms responsible for sharp waves and ripple oscillations.

## Results

To identify the core mechanisms that are responsible for the relationship between learning during exploration and SWRs during rest, we built a scaled-down network model of area CA3 of the rodent hippocampus. In this model, we specifically did not aim to capture all the biological details of hippocampal neurons and circuits; instead, our goal was to reveal and analyze those mechanisms that are essential for the generation of sharp waves, ripple oscillations, and bidirectional activity replay. The main assumptions underlying the construction of our model are listed in Table 1. The complete network consisted of 8000 excitatory pyramidal cells (PCs) and 150 inhibitory parvalbumin-containing basket cells (PVBCs), corresponding roughly to the size of the CA3 area in a 600-μm-thick slice of the mouse hippocampus, which is known to be capable of generating SWRs (*Hájos et al., 2013*; *Schlingloff et al., 2014*). These two cell types are known to be indispensable to the generation of SWRs

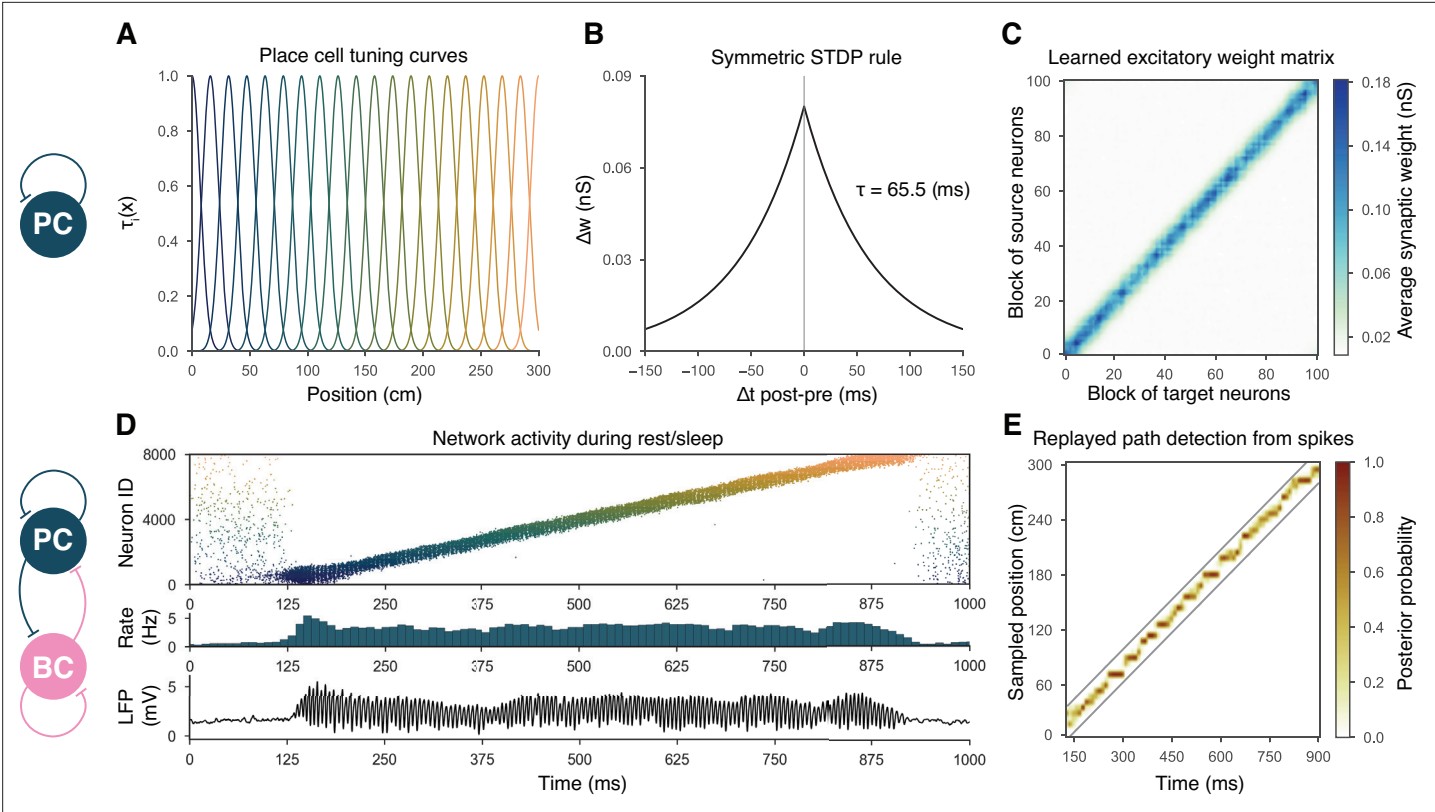

**Figure 1.** Overview of learning and the spontaneous generation of sharp wave-ripples (SWRs) and sequence replay in the model. (**A**) Tuning curves (Equation (1)) of exemplar place cells covering the whole 3-m-long linear track. (**B**) Broad, symmetric spike-timing-dependent plasticity (STDP) kernel used in the learning phase. The time constant was fit directly to experimental data from *Mishra et al., 2016*. (**C**) Learned excitatory recurrent weight matrix. Place cells are ordered according to the location of their place fields; neurons with no place field in the environment are scattered randomly among the place cells. Actual dimensions are 8000 * 8000, but, for better visualization, each pixel shown represents the average of an 80 * 80 square. (**D**) Pyramidal cell (PC) raster plot is shown in the top panel, color-coded, and ordered as the place fields in (**A**). PC population rate (middle), and local field potential (LFP) estimate (bottom panel), corresponding to the same time period. (**E**) Posterior matrix of the decoded positions from spikes within the high activity period shown in (**D**). Gray lines indicate the edges of the decoded, constant velocity path.

The online version of this article includes the following figure supplement(s) for figure 1:

**Figure supplement 1.** The generation of the spike trains of pyramidal cells (PCs) in the exploration phase.

**Figure supplement 2.** Single-cell models.

(*Rácz et al., 2009*; *Ellender et al., 2010*; *English et al., 2014*; *Schlingloff et al., 2014*; *Stark et al., 2014*; *Gulyás and Freund, 2015*; *Buzsáki, 2015*; *Gan et al., 2017*). All neurons in the network were modeled as single-compartment, adaptive exponential integrate-and-fire (AdExpIF) models whose parameters were fit to reproduce in vitro voltage traces of the given cell types in response to a series of step current injections ('Materials and methods' and *Figure 1—figure supplement 2*). Connections of all types (PC-PC, PC-PVBC, PVBC-PC, PVBC-PVBC) were established randomly using connection type-specific probabilities estimated from anatomical studies. All model parameters are detailed in the 'Materials and methods' section and summarized in Tables 2 and 3.

Simulations of the network were run in two phases corresponding to spatial exploration and 'offline' hippocampal states (slow-wave sleep and awake immobility), respectively. During the exploration phase, the spiking activity of PCs was explicitly set to mimic the firing patterns of a population of place cells during simulated runs on a linear track, and the recurrent connections between PCs evolved according to an experimentally constrained, spike-timing-dependent plasticity (STDP) rule (see below). In the subsequent offline phase, we recorded the spontaneous dynamics of the network in the absence of structured external input (or plasticity), analyzed the global dynamics (including average firing rates and oscillations), and looked for the potential appearance of sequential activity

patterns corresponding to the previous activation of place cells during exploration, often referred to as 'replay.' The main features of the model are summarized in *Figure 1*.

## Recurrent weights are learned during exploration via a symmetric STDP rule

During exploration, half of the PCs had randomly assigned, overlapping place fields in the simulated environment, characterized by Gaussian spatial tuning curves, whereas the others fired at low rates in a spatially nonspecific manner ('Materials and methods' and *Figure 1A*). During simulated unidirectional runs along a 3-m-long linear track, these tuning curves, modulated by theta oscillation and phase precession, gave rise to generated spike trains similar to those observed for real place cells under similar conditions ('Materials and methods' and *Figure 1—figure supplement 1*). The simulated spike trains served as inputs for STDP, which was characterized by the broad, symmetric kernel observed in pairs of CA3 PCs recorded in hippocampal slices (*Figure 1B*; *Mishra et al., 2016*). The anatomical connectivity of PCs was sparse and random, assuming 10% PC-PC connection probability (*Lisman, 1999*; *Andersen et al., 2007*), and only these preexisting connections were allowed to evolve ('Materials and methods').

The most prominent feature of the learned recurrent excitatory synaptic weight matrix was its highly organized structure (*Figure 1C*). Relatively few strong (>1 nS) synapses near the diagonal (representing pairs of cells with overlapping place fields) emerged from a background of much weaker connections. Similar symmetric weight structures have been used in continuous 'bump' attractor models of various neural systems such as head direction cells, place cells, and parametric working memory (*Zhang, 1996*; *Samsonovich and McNaughton, 1997*; *Káli and Dayan, 2000*; *Compte et al., 2000*). In these studies, the weights were typically either imposed or learned using rate-based (not STDP) learning rules, and led to stationary patterns of activity in the absence of external input. By contrast, most previous spiking neuron models of sequence learning used temporally asymmetric STDP rules, resulting in a weight structure dominated by feedforward chains (each neuron giving the strongest input to other neurons that follow it in the spatial sequence), and sequential activity patterns that follow these chains of strong connections (*Jahnke et al., 2015*). In order to reveal the combined effect of learned, essentially symmetric connections and realistic cell-type-specific neuronal spike responses, we next explored the spontaneously generated activity patterns in our full network model.

## The dynamics of the network model reproduce key features of SWRs and replay

The sparse recurrent excitatory weight matrix resulting from the phenomenological exploration (*Figure 1C*) was used directly in network simulations mimicking resting periods, in which (with the exception of the simulations shown later in *Figure 2C*) PCs received only spatially and temporally unstructured random synaptic input ('Materials and methods'). The maximal conductances of the other types of connections in the network (the weights of PC-PVBC, PVBC-PC, and PVBC-PVBC connections as well as the weight of the external random input) were optimized using an evolutionary algorithm with network-level objectives, which included physiological PC population firing rates, suppressed gamma (30–100 Hz) oscillation in the PC population, and strong ripple oscillation in the PVBC population (see 'Materials and methods' for details).

Spontaneously generated activity in the network consisted of alternating periods of low activity (mean PC rates below 1 Hz) and high activity (mean PC rates around 3.5 Hz), which resembled the recurring appearance of sharp wave events during slow-wave sleep and quiet wakefulness (*Figure 2A*). Similar to experimental sharp waves, high-activity events in the model were accompanied by transient oscillations in the ripple frequency range (*Figure 2B*; *Buzsáki, 1986*; *Buzsáki et al., 1992*; *Foster and Wilson, 2006*; *Buzsáki, 2015*). We calculated an estimate of the LFP by summing the synaptic inputs of a small, randomly selected subset of PCs (*Mazzoni et al., 2008*), and found that ripple oscillations were reliably present in this signal during the high-activity periods (*Figures 1D and 2B*).

When we plotted the spike times of all PCs with place cells ordered according to the location of their place fields on the track during the learning phase, it became clear that place cell sequences were replayed during simulated SWRs, while activity had no obvious structure in the low-activity periods between SWRs (*Figure 2A*). Interestingly, replay of place cell sequences could occur in either the forward or the backward direction relative to the order of activation in the learning phase

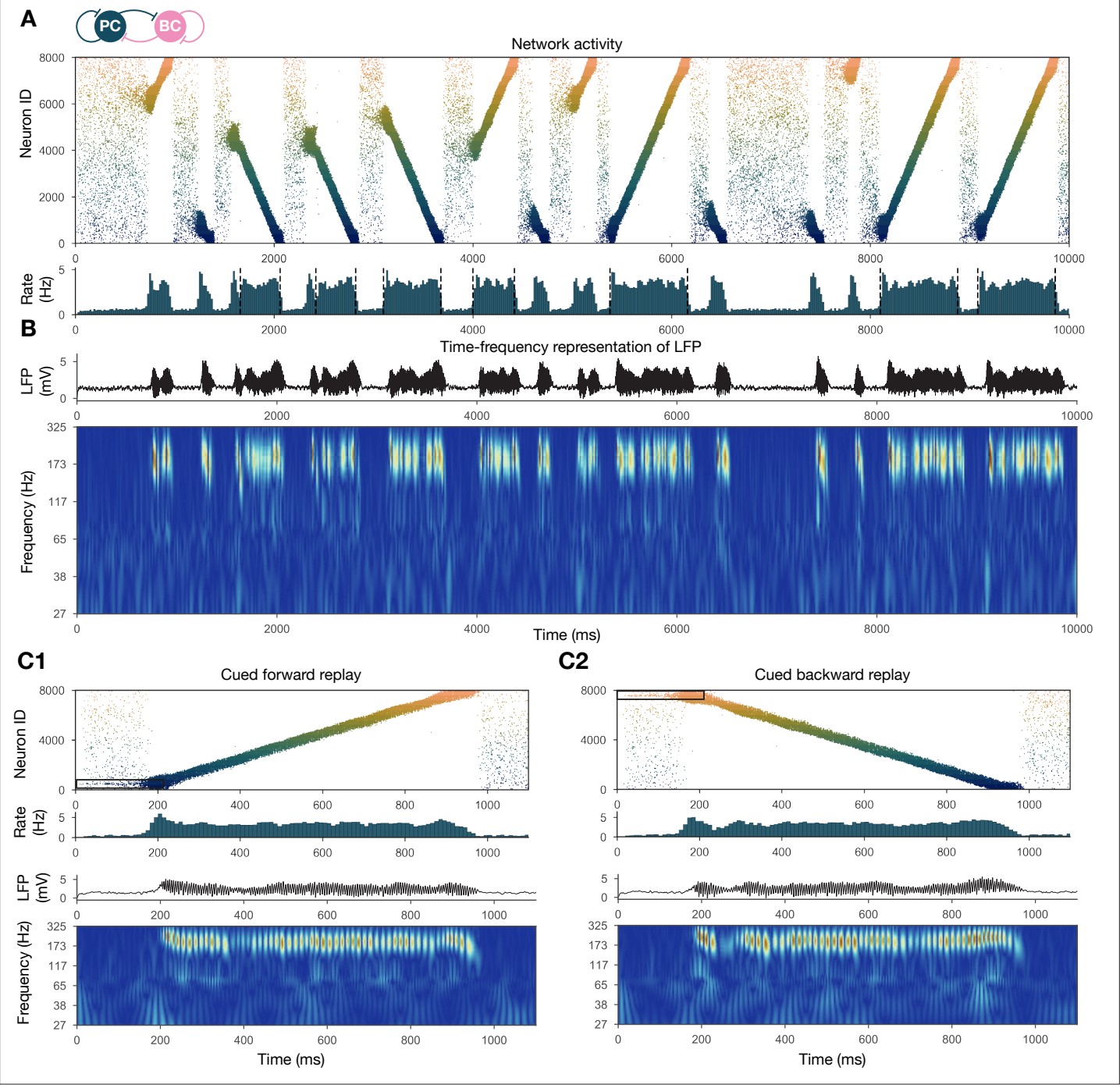

**Figure 2.** Forward and backward replay events, accompanied by ripple oscillations, can occur spontaneously but can also be cued. (**A**) Pyramidal cell (PC) raster plot of a 10-s-long simulation, with sequence replays initiating at random time points and positions and propagating either in forward or backward direction on the top panel. PC population firing rate is plotted below. Dashed vertical black lines indicate the periods marked as sustained high-activity states (above 2 Hz for at least 260 ms), which are submitted to automated spectral and replay analysis. (**B**) Estimated local field potential (LFP) in the top panel and its time-frequency representation (wavelet analysis) below. (**C**) Forward and backward sequence replays resulting from targeted stimulation (using 200-ms-long 20 Hz Poisson spike trains) of selected 100-neuron subgroups, indicated by the black rectangles in the raster plots. (**C1**) Example of cued forward replay. PC raster plot is shown at the top with PC population rate, LFP estimate, and its time-frequency representation (wavelet analysis) below. (**C2**) Same as (**C1**), but different neurons are stimulated at the beginning, leading to backward replay.

The online version of this article includes the following figure supplement(s) for figure 2:

**Figure supplement 1.** The step-size distribution of the decoded paths is much wider than expected.

**Figure supplement 2.** Single-cell characteristics during network simulations.

(*Figure 2A*). Importantly, apart from some initial transients, simultaneous forward and backward replay was never observed. This is because simultaneous replay in both directions induces competition (mediated by global feedback inhibition) between the two active populations, and one of the sequences always dies out after a short period (this is also because mutual support through recurrent excitation diminishes as the two active populations move further away from each other). This is similar to the competitive behavior observed in 'bump attractor' models (*Dayan and Abbott, 2001*). These qualitative observations of sequence replay were confirmed by analyzing high-activity (SWR) periods with a Bayesian place decoding and path fitting method using spatial tuning curves introduced for spike train generation in the exploration part ('Materials and methods' and *Figure 1E*; *Davidson et al., 2009*; *Karlsson and Frank, 2009*; *Olafsdottir et al., 2018*). Interestingly, our network model also reproduced the broad, long-tailed step-size distribution in the sequence of decoded locations during SWRs, as observed experimentally by *Pfeiffer and Foster, 2015*; *Figure 2—figure supplement 1*.

It was also possible to 'cue' replay by giving an additional external stimulation to a small subpopulation ($n = 100$) of PCs that had overlapping place fields in the learned environment. In this case, sequence replay started at the location corresponding to the cells that received extra stimulation (*Figure 2C*). This feature may explain why awake replay tends to be forward when the animal is planning a trajectory from its current location and backward at the goal location.

At the level of single cells, we found that both PCs and PVBCs received approximately balanced excitation and inhibition during SWR events, and inhibitory currents were modulated at the ripple frequency in these periods (*Figure 2—figure supplement 2C and D*). Excitatory inputs dominated between SWRs and during the initiation of SWR events. Only a small minority of PCs fired in individual ripple cycles, while the participation of PVBCs was much higher (but not complete), resulting in a mean PVBC firing rate of ~65 Hz during the SWRs, which is much higher than their baseline rate, but significantly below the ripple frequency (*Figure 2—figure supplement 2A and B*). All of these findings were consistent with experimental results in vitro (*Hájos et al., 2013*; *Schlingloff et al., 2014*) and in vivo (*Varga et al., 2012*; *English et al., 2014*; *Hulse et al., 2016*; *Gan et al., 2017*). The firing of individual PCs appears to differ across experimental conditions, with mostly single spikes in vitro (*Hájos et al., 2013*; *Schlingloff et al., 2014*) and occasional burst firing and a lognormal firing rate distribution in vivo (*Mizuseki and Buzsáki, 2013*). In our simulations, many PCs fired more than one spike per SWR, but these were not bursts in the usual sense as even the shortest ISIs were over 5 ms. The firing rate distribution was skewed, although not particularly long-tailed (*Figure 2—figure supplement 2B and C*).

## SWRs and replay are robust when recurrent excitation is varied

To show that our network model reproduces the wide range of experimental findings presented above in a robust manner, we ran a sensitivity analysis of several parameters. We started by studying the effects of the recurrent excitatory weights, which are the link between exploratory and resting dynamics. To this end, we ran simulations with PC-PC weights that were systematically up- or downscaled after the learning phase, and automatically evaluated various features of the network dynamics, such as population-averaged firing rates, the presence of sequence replay, as well as significant peaks in the ripple frequency range in the power spectra as shown before ('Materials and methods' and *Figure 3*). The network with PC-PC weights multiplied by 0.8 displayed a low-activity, noisy state with severely reduced mean PC firing rate, no sequence replay, and no clear oscillation (*Figure 3A1*). At the 0.9 multiplier level, sequence replays started to appear, but were less frequent than in the baseline model (*Figure 3A2*). As the PC-PC synaptic weights were scaled up, sequence replays became faster (completed in a shorter time) and occurred more often (*Figure 3A3*), a behavior that was stable and realistic up to the 1.5 multiplier level. Ripple oscillations had higher power and appeared at lower multiplier levels in the PVBC population than in the PC population (*Figure 3—figure supplement 1*), suggesting that they originate from the PVBCs and propagate to the PCs, in agreement with theories based on experimental findings (*Buzsáki et al., 1992*; *Ylinen et al., 1995*; *Rácz et al., 2009*; *Schlingloff et al., 2014*; *Stark et al., 2014*; *Gan et al., 2017*). Overall, the qualitative behavior of our model was similar in a fairly wide range of recurrent excitation, although our baseline model was relatively close to the lower end of this range.

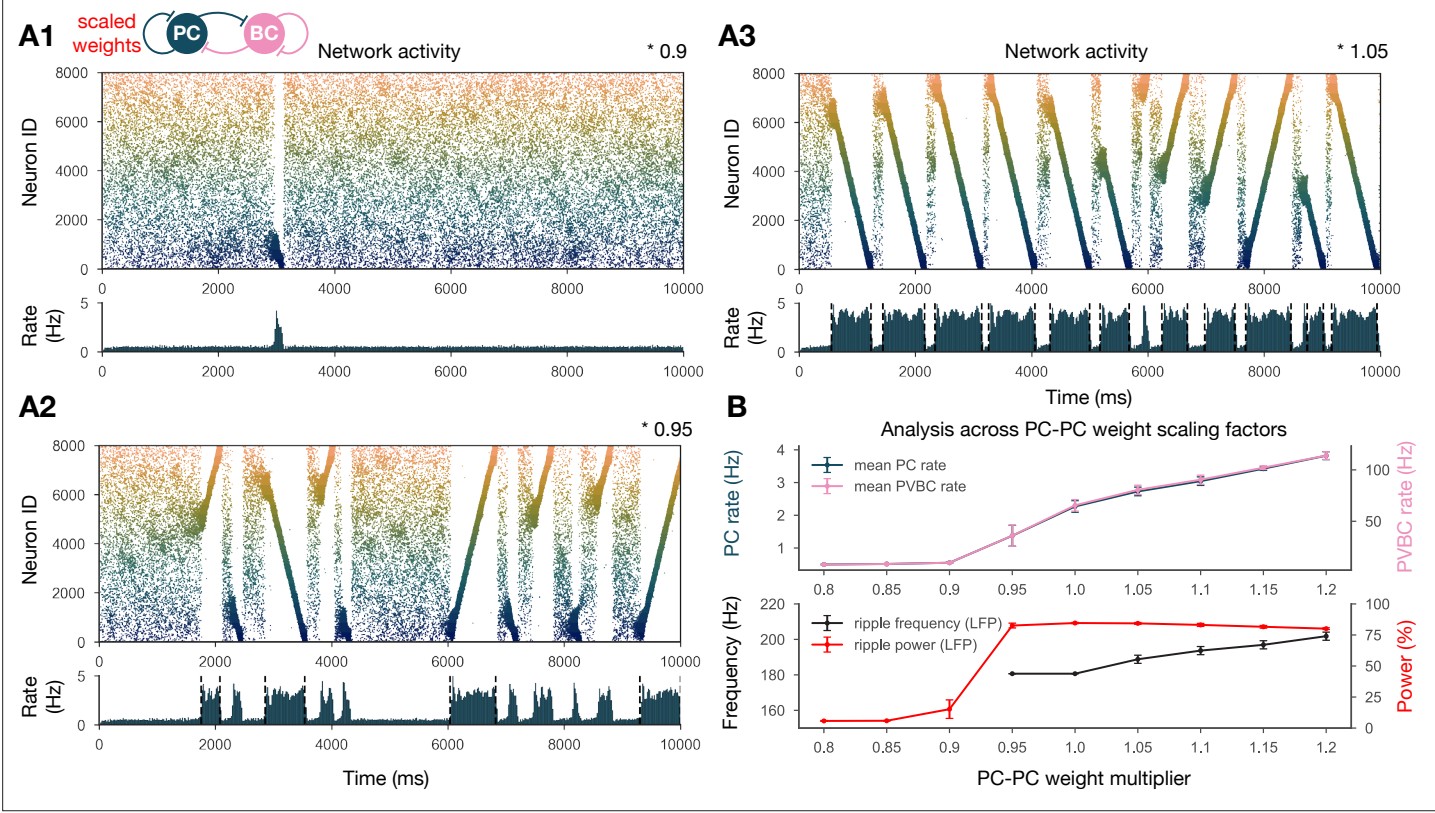

**Figure 3.** Sharp waves, ripple oscillations, and replay are robust with respect to scaling the recurrent excitatory weights. (**A**) Pyramidal cell (PC) raster plots on top and PC population rates at the bottom for E-E scaling factors 0.9 (**A1**), 0.95 (**A2**), and 1.05 (**A3**). (Scaling factor of 1.0 is equivalent to *Figure 2A*.) Dashed vertical black lines have the same meaning as in *Figure 2A*. (**B**) Analysis of selected indicators of network dynamics across different E-E weight scaling factors (0.8–1.2). Mean PC (blue) and parvalbumin-containing basket cell (PVBC) (pink) population rates are shown on top. The frequency of significant ripple oscillations (black) and the percentage of power in the ripple frequency range (red) in the estimated local field potential (LFP) is shown at the bottom. Errors bars indicate standard deviation and are derived from simulations with five different random seeds. See also *Figure 3—figure supplement 1*.

The online version of this article includes the following figure supplement(s) for figure 3:

**Figure supplement 1.** Spectral analysis of network dynamics across different pyramidal cell-pyramidal cell (PC-PC) weight scaling factors.

## Multiple environments can be learned and replayed

Next, we showed that it is possible to store the representations of multiple environments in the weight matrix, and this change does not fundamentally alter the network dynamics (*Figure 4*). In particular, we simulated experience and STDP-based learning in two linear environments with a different but overlapping random set of place cells (*Figure 4A*). The resulting population-level dynamics was quite similar to the one following experience in a single environment, but, during each SWR event, a place cell sequence from either one or the other environment was reactivated (*Figure 4B and C*). Learning two sequences with the same additive symmetric (nondecreasing) STDP rule led to stronger PC-PC synapses on average (*Figure 4A4*), which resulted in a higher overall mean PC rate (*Figure 4B and D*). As a consequence, detectable sequence replays and significant ripple oscillations appeared at lower PC-PC weight multiplier levels (*Figure 4D*).

## Manipulating the model reveals mechanisms of SWR generation and sequence replay

Our network model, equipped with structured recurrent excitation resulting from learning, was able to robustly reproduce recurring sharp wave events accompanied by bidirectional sequence replay and ripple oscillations. Next, we set out to modify this excitatory weight matrix to gain a more causal understanding of how the learned weight pattern is related to the emergent spontaneous dynamics of the network.

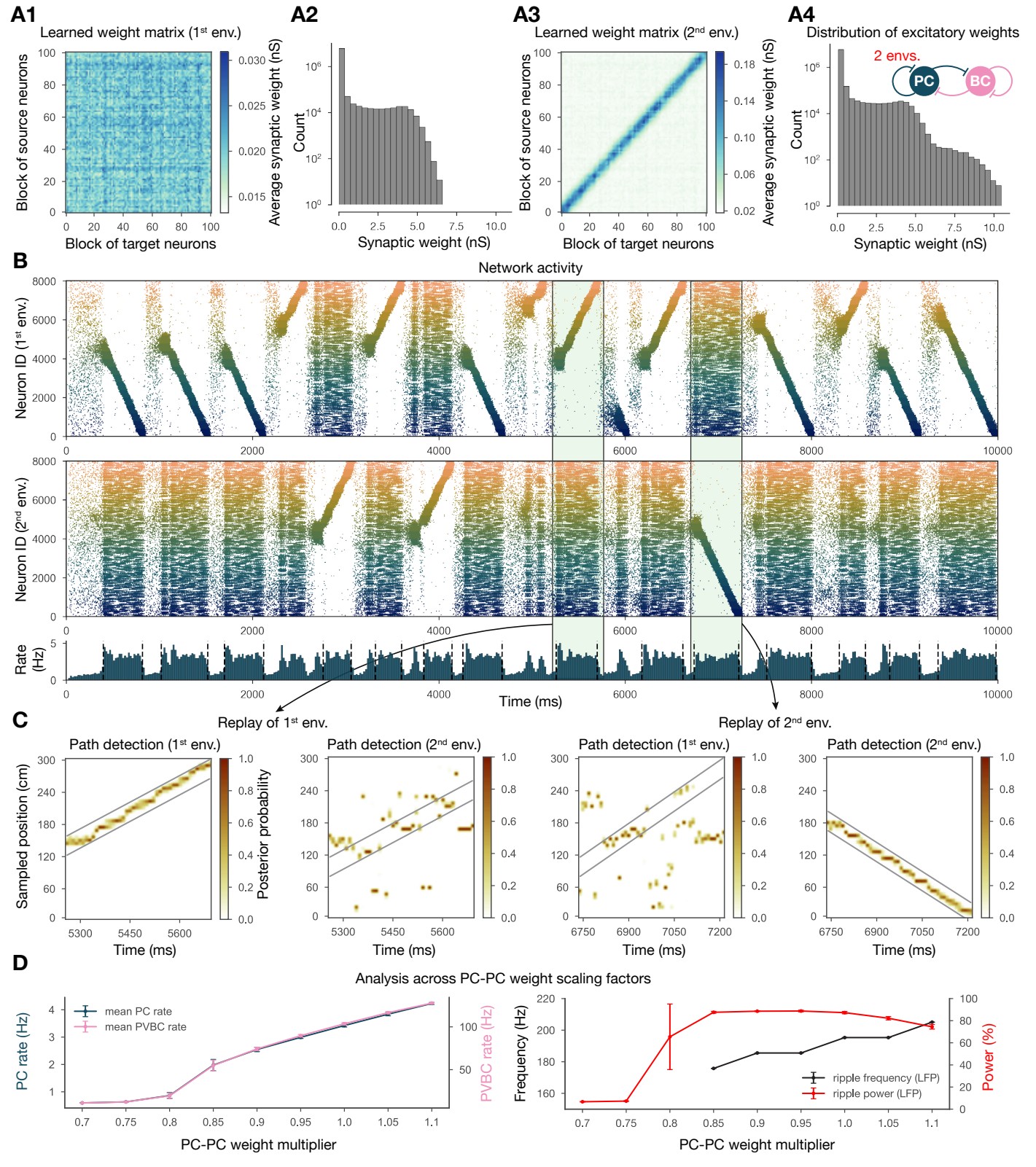

**Figure 4.** Two distinct environments can be learned and replayed by the network. (**A**) Learned excitatory recurrent weight matrices. (**A1**) Weights after learning the first environment. Note that the matrix appears random because neurons are arranged according to their place field location in the second environment, which has not been explored at this point. (**A3**) Weights after learning in the second environment. (**A2, A4**) Distribution of nonzero synaptic weights in the learned weight matrices in (**A1**) and (**A2**), respectively. (**B**) Pyramidal cell (PC) raster plots: in the top panel, neurons are ordered

*Figure 4 continued on next page*

*Figure 4 continued*

and colored according to the first environment; in the middle panel, neurons are ordered and colored according to the second environment; and PC population rate is shown at the bottom (see *Figure 2A*) from a simulation run with 0.9* the modified weight matrix shown in (**A3**). (**C**) Posterior matrices of decoded positions from spikes (see *Figure 1E*) within two selected high-activity periods (8th and 10th from **B**). From left to right: decoding of replay in first environment (eighth event from **B**) according to the first (significant) and second environment; decoding of replay in second environment (10th event from **B**) according to the first and second (significant) environment. (**D**) Analysis of selected network dynamics indicators across different E-E weight scaling factors (0.7–1.1) as in *Figure 3B*.

## Symmetric STDP rule is necessary for bidirectional replay

First, in order to gauge the significance of the experimentally determined symmetric STDP rule, we changed the STDP kernel to the classical asymmetric one that characterizes many other connections in the nervous system (*Figure 5A–C*; *Bi and Poo, 1998*; *Gerstner et al., 2014*). In this case, the learned weight matrix was reminiscent of the feedforward chains that characterized several of the earlier models of hippocampal replay (*Jahnke et al., 2015*; *Chenkov et al., 2017*; *Theodoni et al., 2018*). We found that this weight structure also supported the generation of SWR events and the reactivation

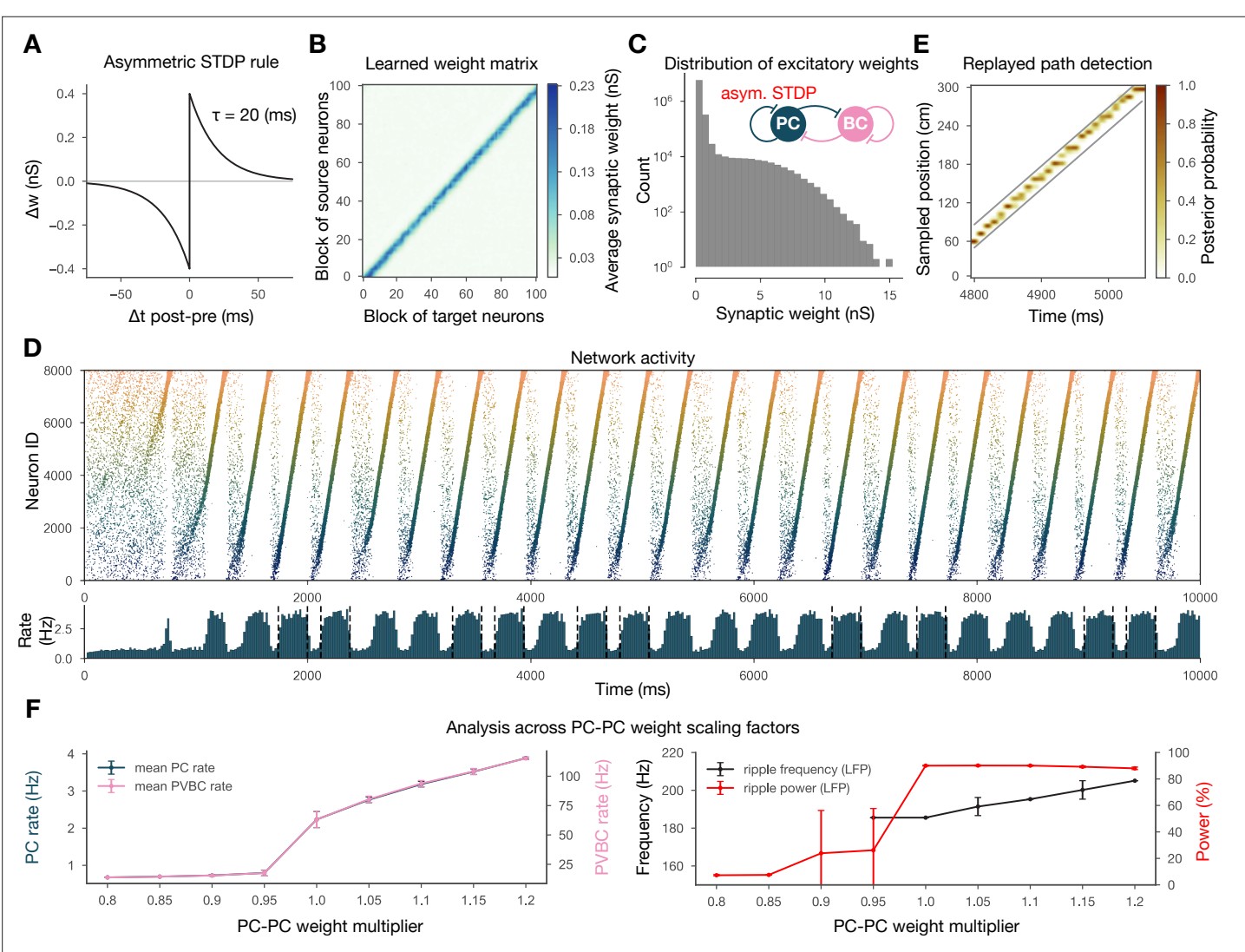

**Figure 5.** Learning with an asymmetric spike-timing-dependent plasticity (STDP) rule leads to the absence of backward replay. (**A**) Asymmetric STDP kernel used in the learning phase. (**B**) Learned excitatory recurrent weight matrix. (**C**) Distribution of nonzero synaptic weights in the weight matrix shown in (**B**). (**D**) Pyramidal cell (PC) raster plot on top and PC population rate at the bottom (see *Figure 2A*) from a simulation run with the weight matrix shown in (**B**). (**E**) Posterior matrix of the decoded positions from spikes (see *Figure 1E*) within a selected high-activity state (sixth one from **D**). (**F**) Analysis of selected network dynamics indicators across different E-E weight scaling factors (0.8–1.2) as in *Figure 3B*.

of learned sequences in our model; however, crucially, sequence replays occurred only in the forward direction (*Figure 5D and E*). *Theodoni et al., 2018* presented a thorough analysis of the relationship between the shape of the (asymmetric) plasticity kernel and sequence replay in a rate-based model, and thus we shifted our focus towards different modifications.

### The structure rather than the statistics of recurrent excitatory weights is critical for SWRs

Inspired by the observation that many network-level properties, such as single PC firing rates, burst index, and participation in SWR events, follow a skewed, lognormal distribution in vivo (*Mizuseki*

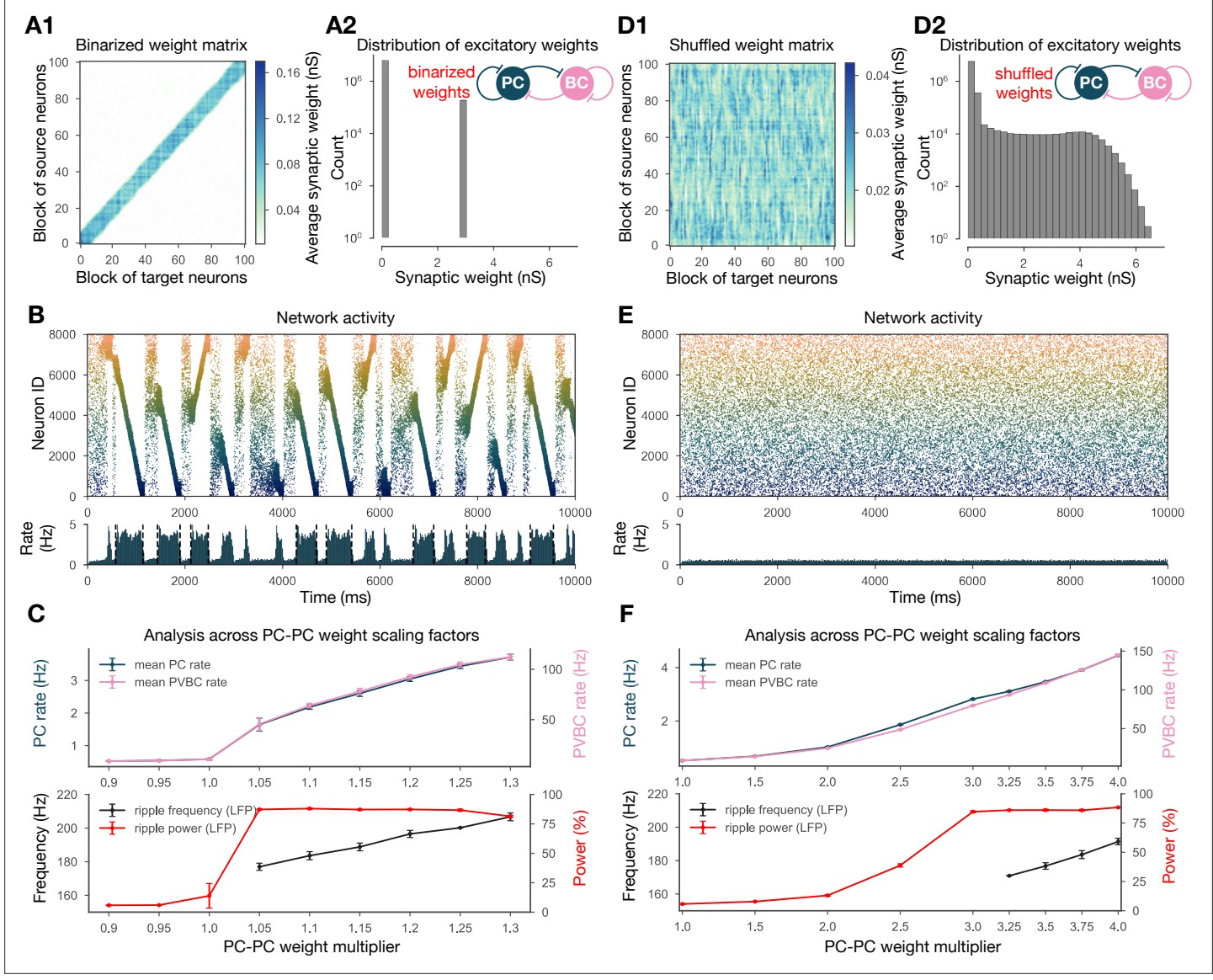

**Figure 6.** Altering the structure of recurrent excitatory interactions changes the network dynamics but altering the weight statistics has little effect. (**A1**) Binarized (largest 3% and remaining 97% nonzero weights averaged separately) recurrent excitatory weight matrix. (Derived from the baseline one shown in *Figure 1C*.) (**A2**) Distribution of nonzero synaptic weights in the learned weight matrix shown in (**A1**). (**B**) Pyramidal cell (PC) raster plot on top and PC population rate at the bottom (see *Figure 2A*) from a simulation run with 1.1* the binarized weight matrix shown in (**A**). (**C**) Analysis of selected network dynamics indicators across different E-E weight scaling factors (0.9–1.3) as in *Figure 3B*. (**D1**) Column-shuffled recurrent excitatory weight matrix. (Derived from the baseline one shown in *Figure 1C*.) (**D2**) Distribution of nonzero synaptic weights in the weight matrix shown in (**D1**) (identical to the distribution of the baseline weight matrix shown in *Figure 1C*). (**E**) PC raster plot on top and PC population rate at the bottom from a simulation run with the shuffled weight matrix shown in (**D1**). (**F**) Analysis of selected network dynamics indicators across different E-E weight scaling factors (1.0–4.0) as in *Figure 3B*. Note the significantly extended horizontal scale compared to other cases.

and Buzsáki, 2013), *Omura et al., 2015* built a network model with recurrent excitatory weights following a lognormal distribution. Their network with unstructured, but lognormally distributed, recurrent synaptic strengths reproduced most of the in vivo observations of *Mizuseki and Buzsáki, 2013*; however, no sequence replay or ripple oscillation was involved. In the network presented here, the distribution of PC-PC weights is the result of the application of STDP to the generated spike trains and does not strictly follow a lognormal distribution, although it has a similar long tail (*Figure 6D2*). In order to establish whether the overall distribution or the fine structure of the weights is the key determinant of neural dynamics in our model, we performed two more drastic perturbations of the recurrent weight matrix itself, starting from the version established in the learning phase of our baseline model (*Figure 1C*).

Our first perturbation kept the structure of the interactions and the overall mean weight intact, but completely changed the distribution of the weights. This was achieved by binarizing the values of the learned weight matrix. Specifically, we divided weights into two groups, the strongest 3% and the remaining 97%, and set each weight in both groups to the group average (*Figure 6A*). Using the modified recurrent synapses between the PCs in the network, we observed extremely similar behavior to our baseline network: recurring sharp waves and sequence replays in both directions, always accompanied by ripple oscillations, with only a small change in the required multiplier for the PC-PC weights (*Figure 6B and C*).

The second modification kept the same overall weight distribution and even the actual values of the outgoing weights for all neurons, but destroyed the global structure of synaptic interactions in the network. To this end, we randomly shuffled the identity of the postsynaptic neurons (by shuffling the columns of the weight matrix). Strong synapses were not clustered anymore along the diagonal (representing interactions between neurons with nearby place fields), but distributed homogeneously within the matrix (*Figure 6D1*). None of the networks equipped with the scaled versions of this shuffled weight matrix exhibited sequence replay, mean PC rates were severely reduced, and no sharp wave-like events were observed (*Figure 6E and F*). On the other hand, with sufficiently amplified (*3.5) PC-PC weights, we detected significant peaks in the ripple frequency range of the LFP (*Figure 6F*).

Taken together, these modifications suggest that, unlike in the model of *Omura et al., 2015*, the distribution of the excitatory recurrent synaptic weights is neither necessary nor sufficient for the observed physiological population activity in our model. In other words, our simulation results suggest that the fine structure of recurrent excitation not only enables coding (sequence replay), but also has a major effect on the global average network dynamics (firing rates, sharp waves, and ripple oscillations) in hippocampal area CA3.

## Cellular adaptation is necessary for replay

As mentioned earlier, most previous models with symmetrical local excitatory interactions and global feedback inhibition functioned as 'bump attractor' networks, in which the dynamics converge to stable patterns of activity involving high rates in a group of neurons with similar tuning properties (adjacent place fields), and suppressed firing in the rest of the excitatory population (*Zhang, 1996*; *Samsonovich and McNaughton, 1997*; *Káli and Dayan, 2000*; *Compte et al., 2000*). Recent work with rate-based models has also shown that these 'stable bumps' can be transformed into 'traveling bumps' by the introduction of short-term depression (*York and van Rossum, 2009*; *Romani and Tsodyks, 2015*; *Theodoni et al., 2018*) or spike threshold adaptation (*Itskov et al., 2011*; *Azizi et al., 2013*). On the other hand, previous spiking models of sequence learning and recall/replay typically relied on temporally asymmetric learning rules and the resulting asymmetric weight matrices to ensure that neurons are reactivated in the same sequence as during learning (*Jahnke et al., 2015*; *Chenkov et al., 2017*), which is also why it is difficult for these models to capture bidirectional replay. *Chenkov et al., 2017* also explored the effects of symmetric connectivity patterns in a spiking network with local feedback inhibition and demonstrated the possibility of bidirectional replay, but the excitatory weights in their models were hard-wired rather than learned. Our model uses the experimentally recorded symmetric STDP rule, which results in symmetrical synaptic interactions (although only at the population level rather than the single-neuron level due to the randomness of connectivity). Since our network generated a bump of activity that traveled unidirectionally in any given replay event rather than a stationary bump, we hypothesized that the cellular-level adaptation that characterized CA3

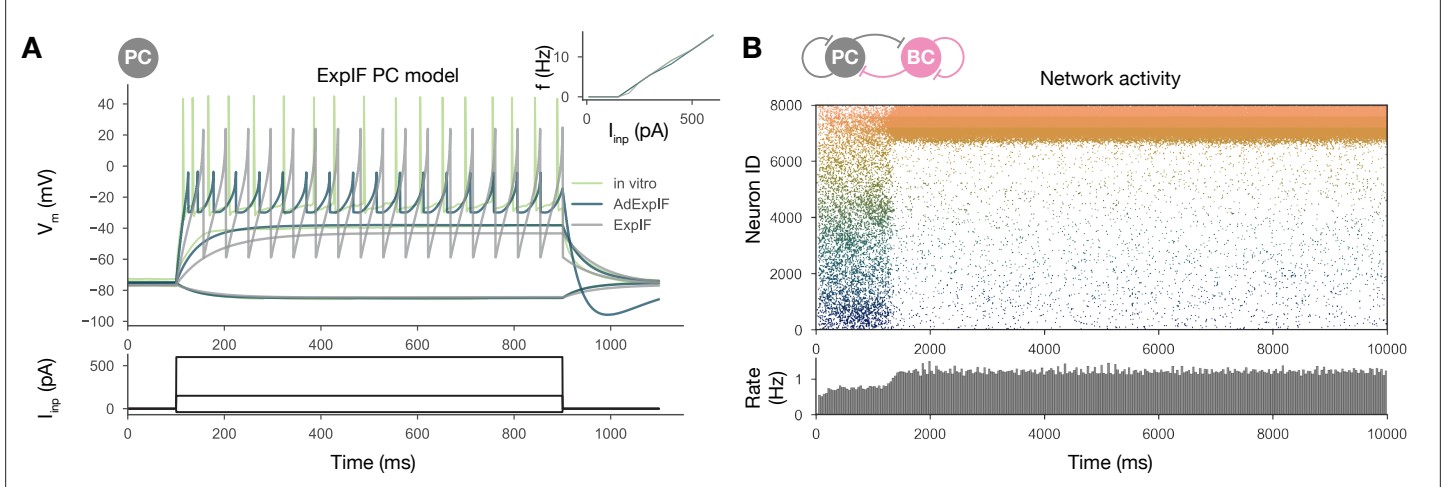

**Figure 7.** Sequential replay requires firing rate adaptation in the pyramidal cell (PC) population. (**A**) Voltage traces of fitted AdExpIF (blue) and ExpIF (gray) PC models and experimental traces (green) are shown in the top panel. Insets show the f–I curves of the in vitro and in silico cells. The amplitudes of the 800-ms-long step current injections shown at the bottom were as follows: –0.04, 0.15, and 0.6 nA. For parameters of the cell models, see *Table 2*. (**B**) PC raster plot of a 10-s-long simulation with the ExpIF PC models, showing stationary activity in the top panel. PC population rate is shown below.

PCs and was also captured by our model may destabilize stable bumps and lead to their constant movement.

To test this hypothesis, we refitted our single-cell data on PC responses to current injections using a modified ExpIF model that did not contain any adaptation mechanism (but was otherwise similar to the baseline model). Although the nonadapting ExpIF model provided a reasonably good fit to our single-cell data (*Figure 7A*), and the weights resulting from learning were identical to the baseline case, the spontaneous network dynamics was completely different: there was no sequence replay for any scaling of the recurrent PC-PC weights; instead, when structured activity emerged from the random background, it was in the form of a stationary rather than a moving bump (*Figure 7B*). Therefore, it was the combination of a symmetric learning rule with cellular adaptation that created the possibility of bidirectional replay in our network model.

## Ripple oscillations are generated by the recurrently coupled inhibitory population

From the weight matrix modifications, we also learned that ripple oscillations can be disentangled from sequence replays and only require sufficient drive to the interconnected PVBC population in our model (*Figure 6F*). The same conclusion was reached by recent optogenetic studies in vitro (*Ellender et al., 2010*; *Schlingloff et al., 2014*) and in vivo (*Stark et al., 2014*). To further investigate the generation of ripples in our model, we simulated and analyzed two additional modified versions of the full network.

First, we disconnected the PVBC network from the PCs and replaced the PC input with independent Poisson synaptic input at rates derived from the firing of the PC population during SWRs in the full simulation (*Figure 8A and B*). In this simplified, recurrently connected, purely PVBC network, we systematically scanned different values of PC input rate and PC-PVBC synaptic weight and measured ripple power as well as the frequency of any significant ripple oscillation as in the full network before (*Figure 8A and B*). We found that ripple oscillations emerged when the net excitatory drive to PVBCs (which is proportional to the product of the incoming weight and the presynaptic firing rate) was sufficiently large. Above this threshold, the properties of the oscillation depended only mildly on the strength of the input (e.g., the frequency of the oscillation increased moderately with increasing drive), and the firing of PVBCs was synchronized mainly by the decay of inhibitory synaptic current evoked by shared input from their peers.

Several experiments have shown that local application of GABA blockers eliminates ripples (*Maier et al., 2003*; *Ellender et al., 2010*; *Schlingloff et al., 2014*; *Stark et al., 2014*); however, in an experimental setup it is hard to distinguish feedback inhibition (PVBC-PC) from reciprocal

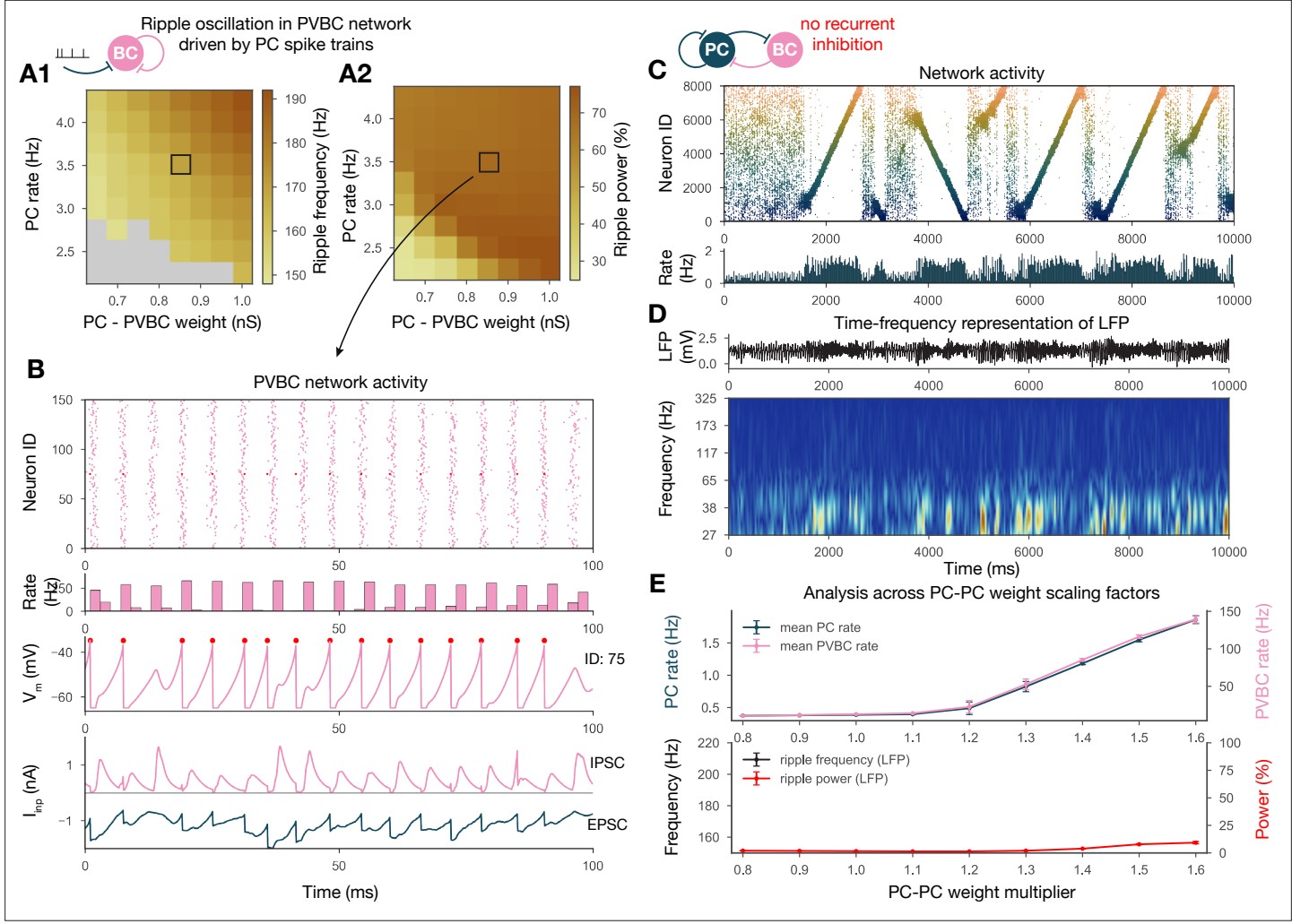

**Figure 8.** Generation of ripple oscillations relies on recurrent connections within the parvalbumin-containing basket cell (PVBC) population. (**A**) Significant ripple frequency (**A1**) and ripple power (**A2**) of a purely PVBC network, driven by (independent) spike trains mimicking pyramidal cell (PC) population activity. Gray color in (**A1**) means no significant ripple peak. (**B**) From top to bottom: raster plot, mean PVBC rate, voltage trace (of a selected cell), EPSCs, and IPSCs of the selected cell from the middle (100-ms-long window) of a simulation used for (**A**). Ripple frequency and power corresponding to this simulation are marked with a black rectangle in (**A1**) and (**A2**). (**C**) PC raster plot on top and PC population rate at the bottom from a simulation run with a network with 1.3* baseline E-E weight matrix but without any PVBC-PVBC synapses, featuring stochastic forward and backward replays but no ripple oscillation (see below). (**D**) Estimated local field potential (LFP) in the top panel and its time-frequency representation (wavelet analysis) below. (Compared to *Figure 2B*, there is increased power in the gamma range, but no ripple frequency oscillation.) (**E**) Analysis of selected network dynamics indicators across different E-E weight scaling factors (0.8–1.6) as in *Figure 3B*.

inhibition (PVBC-PVBC). As a final perturbation, we modified the full baseline model by eliminating only the recurrent inhibitory synapses (*Figure 8C and D*). The resulting dynamics were stable and with enhanced (*1.3) PC-PC weights it also displayed sequence replay, but ripple oscillations were never observed (*Figure 8D and E*). Taken together, these results support the conclusions of previous modeling (*Brunel and Wang, 2003*; *Geisler et al., 2005*; *Taxidis et al., 2012*; *Donoso et al., 2018*; *Ramirez-Villegas et al., 2018*) as well as experimental studies (*Buzsáki et al., 1992*; *Ylinen et al., 1995*; *Rácz et al., 2009*; *Ellender et al., 2010*; *Schlingloff et al., 2014*; *Stark et al., 2014*; *Gulyás and Freund, 2015*; *Gan et al., 2017*), proposing that ripple oscillations are generated in strongly driven, recurrently connected inhibitory networks by the fast inhibitory neuronal oscillation (FINO) mechanism. In fact, recurrent inhibitory connections were both necessary and sufficient for the generation of ripple oscillations in our model.

## Discussion

Using a data-driven network model of area CA3 of the hippocampus that reproduces the main characteristics of SWRs, we examined the link between learning during exploration and the network dynamics in resting periods. Our principal findings from analyzing and manipulating this model are as follows: (1) structured (learned) recurrent excitation in the CA3 region not only enables coding and memory, but is critical for the generation of SWRs as well; (2) the symmetric STDP rule described by *Mishra et al., 2016*, in combination with cellular adaptation in CA3 PCs, provides an explanation for the coexistence of forward and reverse replays; (3) the pattern of strong connections in the network rather than the overall weight statistics may be critical for the emergence and key properties of SWRs and replay in area CA3; and (4) ripple oscillations are generated in the strongly driven, recurrently connected network of fast-spiking PVBCs by the FINO mechanism (*Schlingloff et al., 2014*) (also known as PYR-INT-INT [*Stark et al., 2014*; *Buzsáki, 2015*; *Ramirez-Villegas et al., 2018*]).

### Connections of sharp waves, sequence replay, and ripple oscillations

SWRs represent the most synchronous physiological activity pattern with the largest excitatory gain in the mammalian brain (*Buzsáki et al., 1983*; *Buzsáki et al., 1992*; *Buzsáki, 1989*; *Buzsáki, 2015*). Under normal conditions, ripples are typically observed riding on top of naturally emerging sharp waves. More recently, using optogenetics, *Schlingloff et al., 2014* and *Stark et al., 2014* managed to decouple ripples from sharp waves by directly and indirectly activating the interconnected network of PVBCs. Our in silico results perfectly parallel this work: without drastic, nonphysiological modifications of the model ripples were always tied to sequence replay, which was in turn associated with bouts of increased spiking activity in the network (the sharp waves). When we separated the PVBC network, we found that a relatively high (>2 Hz) mean firing rate in the PC population was required for inducing ripple oscillation, a condition that was satisfied only during sharp wave events and the associated sequence replay in our full baseline network. When the PC population reaches this frequency after a stochastically initiated buildup period (*Schlingloff et al., 2014*), the strongly driven, high-frequency firing of PVBCs is synchronized and phase-locked via reciprocal inhibition. Thus, the learned recurrent PC-PC synaptic weights are responsible for coding, govern sequence replay, and, by giving rise to high PC activity during the replay, they also cause the ripples. In summary, memory storage and recall, as well as the main hippocampal oscillations and transient activity patterns, are intimately interconnected in our unifying model.

### Experimental predictions

The identification of the mechanisms in the model through which specific synaptic interactions, plasticity, and cellular properties give rise to sequence generation and particular aspects of population dynamics (sharp waves and ripple oscillations) also allows us to make explicit experimental predictions. Although some of the manipulations that we used to probe our model (such as shuffling the weights) are not feasible in the biological system, others could probably be replicated using modern (e.g., optogenetic or chemogenetic) tools. As an important example, our model predicts that selectively blocking recurrent inhibitory connections between PVBCs should completely eliminate ripple oscillations, while having only minor quantitative effects (through modifying the effective gain of feedback inhibition of PCs) on sharp waves and sequence replay. By contrast, blocking or substantially attenuating recurrent excitatory interactions in area CA3 is expected to simultaneously eliminate sharp waves, ripple oscillations, and autonomous sequence replay.

Recent experimental findings indicated that the content of replay may be substantially different depending on the experience of the animal (*Pfeiffer and Foster, 2015*; *Stella et al., 2019*; *Kay et al., 2020*). In our model, the nature of replayed sequences depends mostly on the activity patterns of hippocampal neurons during exploration, which determine, through learning, the subsequent pattern of recurrent interactions. In the basic scenario explored in our paper, learning occurred during repeated runs along a single linear track, which led to reactivations consisting of random chunks of the single learned trajectory, and allowed our model to replicate the replay dynamics observed by *Pfeiffer and Foster, 2015*. However, other types of learning experience would be expected to result in different reactivation dynamics. For instance, random exploration in an open field, as investigated by *Stella et al., 2019*, may be expected to result in a 2D bump attractor structure encoded in the recurrent weights (similarly to the earlier model of *Káli and Dayan, 2000*), and this, combined with

adaptation and some random noise, may explain the diffusive nature of reactivation in their experiments. Similarly, the learning of alternative routes in the experiments of *Kay et al., 2020*, combined with adaptation, may explain the spontaneous switching between representations of alternative future paths. Overall, a general prediction of our framework is that one could deduce the nature of reactivations in different experiments using the same model simply by changing the spike statistics during the learning phase so that it mimics the corresponding experimental observations. It also follows from this argument that the different results of these studies should stem primarily from the different environments and protocols used during training rather than, for instance, differences in analysis methods.

Experimental studies focusing on conventional forms of sequence reactivation found that the dominant direction of replay depended on the behavioral state of the animal (*Foster and Wilson, 2006*; *Diba and Buzsáki, 2007*; *Pfeiffer and Foster, 2013*), and replay often started at behaviorally relevant locations (*Karlsson and Frank, 2009*). In our model, the starting point and direction of replay depend on systematic biases and random variations in the weight structure, but also on (random or deterministic) spatio-temporal structure in the external inputs. We believe that the cued replay scenario that we simulated is not entirely artificial – even real environments may contain choice points where the animal pauses to plan its route, or reward sites that again make the animal stop and may also activate subcortical modulatory systems that affect plasticity. At intermediate sites, systematic bias could be introduced during exploration by factors that influence behavior (e.g., running speed), neuronal activity or plasticity (e.g., attention), especially if such factors can change the fully symmetric nature of the synaptic plasticity rule. Random variations in connectivity or the weights (e.g., due to stochastic spiking during learning) will lead to a location-dependent systematic bias towards forward or reverse replay. Finally, (randomly occurring or input-driven) sequences in the external input are also expected to bias the direction of replay even at intermediate points. Although a detailed analysis of the relative impact of these factors will require further efforts, we can already make some predictions regarding the effects of various experimental manipulations. In particular, it should be possible to influence the starting point and direction of replay by the selective activation of place cells representing a particular location (or a sequence of locations) in offline states, although the rapid sequential activation of functionally defined chains of neurons might be technically challenging. More importantly, our model also predicts that similar manipulations of neuronal activity *during exploration* should also have an impact on the content of replay, and even on the global properties of sharp waves, through their influence on the learned synaptic weight structure.

Finally, our modeling results also suggest that it would be critical to explore whether the unusual, temporally symmetric STDP rule uncovered by *Mishra et al., 2016* correctly describes the plasticity of CA3 excitatory recurrent collaterals under all physiological circumstances, or perhaps the rule is different (not entirely symmetric or even antisymmetric) under some conditions in vivo, for example,

**Table 1.** List of modeling assumptions.

| | |
|---|---|
| 1 | In the absence of unified datasets, it was assumed that published parameters from different animals (mouse/rat, strain, sex, age) can be used together to build a general model. |
| 2 | Connection probabilities were assumed to depend only on the presynaptic cell type and to be independent of distance. |
| 3 | Each pyramidal cell was assumed to have a place field in any given environment with a probability of 50%. For simplicity, multiple place fields were not allowed. |
| 4 | When constructing the 'teaching spike trains' during simulated exploration, place fields were assumed to have a uniform size, tuning curve shape, and maximum firing rate. |
| 5 | For simplicity, all synaptic interactions in the network were modeled as deterministic conductance changes. Short-term plasticity was not included, and long-term plasticity was assumed to operate only in the learning phase. |
| 6 | When considering the nonspecific drive to the network in the offline state, it was assumed that the external input can be modeled as uncorrelated random spike trains (one per cell) activating strong synapses (representing the mossy fibers) in the pyramidal cell population. |
| 7 | Some fundamental assumptions are inherited from common practices in computational neuroscience; these include modeling spike trains as Poisson processes, capturing weight changes with additive spike-timing-dependent plasticity, describing cells with single-compartmental AdExpIF models, modeling a neuronal population with replicas of a single model, and representing synapses with conductance-based models with biexponential kinetics. |
| 8 | When comparing our model to in vivo data, an implicit assumption was that the behavior of a simplified model based on slice constraints can generalize to the observed behavior of the full CA3 region in vivo, in the context of studying the link between activity-dependent plasticity and network dynamics. |

depending on behavioral variables and the associated pattern of subcortical inputs. One specific prediction of our model is that any factors that change the symmetric nature of the plasticity rule during exploration should alter the balance of forward and backward replay events during subsequent SWRs.

## Biological plausibility of the model

The network model presented here was constrained strongly by the available experimental data. Many cellular and synaptic parameters were fit directly to in vitro measurements, and most functional parameters correspond to in vivo recordings of hippocampal place cells. Nevertheless, there are certainly many biological features that are currently missing from our model. We do not see this as a major limitation of our study as our goal was to provide a mechanistic explanation for a core set of phenomena by identifying the key underlying biological components and interactions. On the other hand, our model can be systematically refined and extended, especially when new experimental data become available. The main assumptions we made when constructing the model are explicitly stated in *Table 1*. Here, we briefly discuss some of these assumptions, as well as some remaining discrepancies between our simulation results and the corresponding experimental data.

The 10% PC-PC connection probability is based on the classical viewpoint that considers the CA3 region as a highly interconnected network (*Lisman, 1999*; *Andersen et al., 2007*). Although a recent physiological study (*Guzman et al., 2016*) estimated <1% connection probability in 400-μm-thick slices, the authors concluded from 'virtual slicing' experiments that recurrent axons were substantially reduced. This is in agreement with *Li et al., 1994*, who reported at least 70% axonal loss in 400 μm slices. Thus, the in vivo connection probability is likely to be considerably higher than 1%. Furthermore, due to the long-tailed distribution of synaptic weights, many of the anatomically existing connections that are represented in our model may be too weak to be reliably detected in paired recording experiments.

Our model network contained 8000 PCs and 150 PVBCs, which is rather small compared to the full rodent CA3 region. While in vitro studies suggest that this network size is sufficient for the generation of SWRs, and that these two cell types are the key players in this process, it is likely that the much larger network size and the various additional cell types modify the quantitative aspects of SWRs in vivo.

For simplicity, we also decided not to model all the different types of variation in single-cell and synaptic properties that characterize real cortical networks. However, it is important to note that several sources of variability (e.g., those resulting from the random assignment of place field locations, sparse, randomized connectivity, and stochastic, independent external drive) were still included in the model. The variability that we include is sufficient to avoid the regular, unnaturally synchronous dynamics that often characterize homogeneous networks, and results in variable, natural-looking spike patterns during both exploration and spontaneous SWR activity. Including additional forms of heterogeneity would likely increase the cell-to-cell variability of activity (such as firing rates), but (up to a certain limit) would not be expected to change our results qualitatively.

We modeled activity-dependent synaptic plasticity based on the study of *Mishra et al., 2016*, who uncovered an unusual, temporally symmetric form of STDP in the recurrent excitatory connections between CA3 pyramidal neurons. We fit the temporal kernel of our STDP rule directly to their data, assumed linear summation of synaptic modifications from all the spike pairs that occurred during the exploration phase, and ignored all higher-order (triplet, etc.) spike interactions. However, we note that the details of the plasticity rule are not expected to have a major impact on our results concerning the emergent dynamics of the CA3 network during resting periods as long as learning results in the strengthening of connections between PCs that are approximately co-active during exploration, independent of the exact temporal order of their spikes. This requirement would be satisfied by a variety of different learning rules, including the behavioral timescale plasticity rule recently described in hippocampal CA1 PCs (*Bittner et al., 2017*).

The complete absence of plasticity in the second phase of our simulations may not be realistic. On the other hand, plasticity is known to be strongly modulated by brain state (e.g., through the action of subcortical inputs on hippocampal cells and synapses), so plasticity may be substantially weaker or different in offline states (such as slow-wave sleep; see *Hasselmo, 2006*; *Fuenzalida et al., 2021* for reviews). Finally, since replay recapitulates activity correlations in the learning phase, the main effect

of plasticity during replay would be to reinforce existing patterns of weights and activity, although some kind of homeostatic mechanism may be required to prevent uncontrolled positive feedback in this case.

One substantial difference between SWRs in the model and those recorded in vivo is the duration of the SWR events and the associated replay. Ripple episodes typically last 40–100 ms in vivo (*O'Keefe and Nadel, 1978*; *Buzsáki et al., 1983*; *Buzsáki et al., 1992*; *Ylinen et al., 1995*), although *Fernández-Ruiz et al., 2019* recently showed that learning in spatial memory tasks is associated with prolonged SWRs and replays. In our model, SWRs can be up to 800 ms in duration as they are terminated when the replayed sequence reaches either the end or the beginning of the learned trajectory (depending on the direction of replay), thus the length of the track determines the maximal duration of the SWR, in combination with the speed of replay (i.e., the rate at which activation propagates across the population of place cells). The speed of replay in the model is consistent with experimental findings, but a single replay event can cover the whole 3-m-long track. *Davidson et al., 2009* also used a long track in their experiments; however, they reported that the whole path was never replayed during a single SWR, only short segments, which combined to cover the whole track over multiple SWRs. Therefore, it appears likely that our simplified model lacks some additional mechanisms that contribute to the termination of individual SWRs in the experiments. For example, building on the work of *York and van Rossum, 2009*, some rate-based models of sequence replay in a circular environment (*Romani and Tsodyks, 2015*; *Theodoni et al., 2018*) included short-term synaptic depression as a mechanism for terminating replay. However, *Guzman et al., 2016* found pseudo-linear short-term plasticity profiles for the recurrent PC-PC connections at physiological temperatures (although depression was present in recordings at room temperature). Another possibility is that an additional cell type that is not currently included in our model is responsible for the termination of SWR events. More specifically, the duration of the in silico SWRs could probably be controlled by a second interneuron population providing delayed, long-lasting feedback inhibition to the PCs. Three-population models were recently analyzed in detail by *Evangelista et al., 2020*, who found that competitive interactions between PVBCs and another (so far unidentified) interneuron population inhibiting PCs, combined with appropriate short-term synaptic plasticity of the relevant connections, could explain many key experimental findings regarding the initiation and termination of SWRs.

Finally, we have presented our work as a model of the hippocampal CA3 area. This is because this area is known to be able to generate SWRs on its own, has modifiable recurrent connections that are thought to be essential for memory, and there are sufficient experimental data from this region to constrain the model. However, other cortical regions such as CA2, the subiculum, and entorhinal cortex are likely involved in the initiation of SWRs under normal conditions (*Oliva et al., 2016*; *Oliva et al., 2020*), and the mechanisms we describe here may be operational in these and other brain areas as well.

## Previous models of SWRs and sequence replay

In this study, our main objective was to build a simplified, yet powerful model of area CA3 that is constrained by experimental data at all (possible) levels, and thus allows us to uncover the mechanistic links between learning, neural population dynamics, and the representation of spatial (or other) sequences in the hippocampus during different behavioral states. Although there is a plethora of hippocampal models that shed light on some of these aspects (these models have been recently reviewed [*Buzsáki, 2015*; *Jahnke et al., 2015*] and are also cited when relevant throughout the 'Results' section), there are only a handful of recent models that attempted to treat all of them within a single coherent framework.

The study of *Jahnke et al., 2015* is probably the most similar in spirit to ours as it also explores the relationship between learning, replay, and SWRs. One distinguishing feature of their model is that it relies on the nonlinear amplification of synaptic inputs by dendritic spikes in CA3 PCs for the generation of both sequential activity and ripple oscillations (*Memmesheimer, 2010*). Importantly, replay always occurs in the forward direction in their model as it relies on feedforward chains of strong weights in the network, established by an asymmetric STDP rule that is quite distinct from the one that was later found empirically by *Mishra et al., 2016*. In addition, the generation of ripple oscillations in their model relies on synchronized pulses of activity generated by interconnected PCs, while recent experimental findings appear to provide causal evidence for the involvement of fast-spiking PVBCs in

ripple frequency oscillation generation (*Rácz et al., 2009*; *Ellender et al., 2010*; *English et al., 2014*; *Schlingloff et al., 2014*; *Stark et al., 2014*; *Gulyás and Freund, 2015*; *Buzsáki, 2015*; *Gan et al., 2017*). Finally, SWRs need to be evoked by synchronous external input in the model of *Jahnke et al., 2015*, while they can also emerge spontaneously in our model.

*Malerba and Bazhenov, 2019* developed a combined model of areas CA3 and CA1 to study the generation of sharp waves in CA3 and associated ripple oscillations in CA1. This model relies on distance-dependent connection probabilities in the network for the generation of spatially localized SWR events. The study shows that modifying the recurrent excitatory weights via an asymmetric STDP rule during a simulated learning epoch biases the content of SWRs towards the (forward) reactivation of learned trajectories. Ripple oscillations are modeled only in CA1 and, in contrast to our model (and the models of *Taxidis et al., 2012*; *Donoso et al., 2018*; *Ramirez-Villegas et al., 2018*), their generation is independent of recurrent inhibition.

A notable recent example of functionally motivated (top-down) modeling of these phenomena is the study of *Nicola and Clopath, 2019*. The authors designed and trained (using supervised learning methods) a network of spiking neurons to generate activity sequences that were either tied to the population-level theta oscillation or occurred spontaneously in isolation (in a compressed manner) depending on the presence or absence of an external theta-frequency input. Interestingly, these results were achieved by tuning specifically the inhibitory weights in the network, while all other models (including ours) rely on plasticity in the recurrent excitatory synapses. Their model produced forward replay of sequences by default; however, sequences could be reversed by the activation of a distinct, dedicated class of interneurons.

To our best knowledge, ours is the first model that autonomously generates SWRs and replay in a spiking network model using synaptic weights established via the experimentally observed symmetric STDP rule. The model of *Haga and Fukai, 2018* used symmetric STDP (in combination with short-term plasticity) to modify an existing (pre-wired) weight structure, and showed that these changes biased evoked activity sequences towards reverse replay. Neither spontaneous sharp waves nor ripple oscillations were observed in this model.

We believe that our approach of fitting the parameters of our single-cell models directly to experimental data to mimic the physiological spiking behavior of real PCs and PVBCs is also quite unique. This enabled our models of PCs to capture spike frequency adaptation, which proved to be essential for the generation of propagating activity (sequence replay) despite the essentially symmetric nature of synaptic interactions.

## Conclusions

At a more general level, our results highlight the significance of some previously neglected interactions between three fundamental components of brain function: population dynamics, coding, and plasticity. Specifically, the different types of population dynamics (including oscillations and transients such as sharp waves) are mostly seen as a background over which coding and plasticity occur, while the impacts of plasticity and especially neural representations on the generation of population activity patterns are rarely considered. However, our results strongly suggest that the structured network interactions resulting from activity-dependent learning (or development) lead to specific spatio-temporal activity patterns (such as autonomously generated replay sequences), which are in turn critical for the emergence of physiological population activity (such as sharp waves and ripple oscillations). Therefore, our study indicates that the complex structure of synaptic interactions in neuronal networks may have a hitherto unappreciated degree of control over the general mode of activity in the network and should be taken into account by future theories and models of population activity patterns in any part of the nervous system.

## Materials and methods

In order to investigate the mechanisms underlying hippocampal network dynamics and how these are affected by learning, we built a simplified network model of area CA3. This scaled-down version of CA3 contained 8000 PCs and 150 PVBCs, which is approximately equivalent to cell numbers in area CA3 in a 600-μm-thick hippocampal slice based on our previous estimates (*Schlingloff et al., 2014*), which are also in good agreement with other estimates (*Bezaire and Soltesz, 2013*; *Donoso et al.,*

*2018*; *Evangelista et al., 2020*). The connection probability was estimated to be 25% for PVBCs (*Schlingloff et al., 2014*) and 10% for PCs (Table 3), and was independent of distance in the model. Half of the PCs were assumed to have place fields on the simulated 3-m-long linear track.

Only PCs received external input; during exploration, they were activated directly to model place-associated firing; otherwise, they received synaptic input (through the mossy fibers) in the form of uncorrelated Poisson spike trains with a mean rate of 15 Hz. As the hallmark of granule cell activity in the dentate gyrus is sparse, low-frequency firing (*Jung and McNaughton, 1993*; *Skaggs et al., 1996*), and each CA3 PC is contacted by only a few mossy fibers, the physiological mean rate of mossy fiber input to PCs is probably substantially lower (especially in offline states of the hippocampus). On the other hand, real CA3 PCs also receive direct input from the entorhinal cortex, and also spontaneous EPSPs from a large population of recurrent collateral synapses. Overall, these other numerous, but small-amplitude inputs may be responsible for most of the depolarization required to bring CA3 PCs close to their firing threshold, in which case mossy fiber input at a significantly lower rate would be sufficient to evoke the same number of action potentials in CA3 PCs.

Network simulations were run in Brian2 (*Stimberg et al., 2019*). Learning of the structure of recurrent excitation, single-cell and network optimization, and the analysis of the network simulations are detailed in the following sections. A comprehensive list of assumptions made during the model building process is presented in *Table 1*.

## Spike trains during exploration

Spike trains mimicking CA3 PC activity during exploration were generated with exponentially distributed interspike intervals (ISIs) with mean $1/\lambda$, giving rise to Poisson processes. Spike trains of nonplace cells had constant mean firing rates of $\lambda = 0.1$ Hz. For the spike trains of the randomly selected 4000 place cells, homogeneous Poisson processes with $\lambda = 20$ Hz were generated, and spike times were accept-reject sampled with acceptance probability coming from place cell-like tuning curves (Equation (2)), which led to inhomogeneous Poisson processes with time-dependent rate $\lambda(t)$. Tuning curves were modeled as Gaussians centered at randomly distributed positions and standard deviation ($\sigma$) set to cover 10% of the 3-m-long linear track (Equation (1)). Edges of the place fields were defined where the firing rate dropped to 10% of the maximal 20 Hz (*Dragoi and Buzsáki, 2006*). Tuning curves were also modulated in time by the background $f_\theta = 7$ Hz theta activity, and phase precessed up to 180° at the beginning of the place field (*O'Keefe and Recce, 1993*). The firing rate of the th place cell was calculated as follows:

$$\tau_i(x) = \exp\left(\frac{-(x-m_i^{PF})^2}{2\sigma^2}\right) \tag{1}$$

$$\lambda_i(t) = \lambda_{max} \times \tau_i(x(t)) \times \cos\left(2\pi f_\theta t + \frac{\pi}{l^{PF}}\left(x(t) - s_i^{PF}\right)\right) \tag{2}$$

where $\tau_i(x)$ is the spatial tuning curve of the th neuron, $x(t)$ is the position of the animal, $m_i^{PF}$, $l^{PF} = 0.3$ m, and $s_i^{PF}$ are the middle, length, and start of the given place field, respectively; $\lambda_{max} = 20$ Hz is the maximum in-field firing rate.

Spikes within a 5 ms refractory period of the previous spike were always rejected. The speed of the animal was set to 32.5 cm/s, thus each run took ~9.2 s, after which the animal was immediately 'teleported back' to the start of the linear track. Generated spike trains were 400 s long, leading to ~43 repetitions on the same linear track.

## Learning via STDP

STDP was implemented by an additive pair-based learning rule, evaluated at spike arrivals (*Kempter et al., 1999*; *Gerstner et al., 2014*). Synaptic weights evolved as follows:

$$\Delta w_+ = A_+ \exp\left(-\frac{\Delta t}{\tau_+}\right) \text{ at } t_{post} \text{ if } t_{pre} < t_{post} \tag{3}$$

$$\Delta w_- = A_- \exp\left(\frac{\Delta t}{\tau_-}\right) \text{ at } t_{pre} \text{ if } t_{pre} > t_{post} \tag{4}$$

where $\Delta t = t_{post} - t_{pre}$ is the time difference between action potentials, and $A_\pm$ describes the weight update, which decayed exponentially with time constants $\tau_\pm$. Synaptic weights were cropped

at $w_{max} = 20$ nS. To reproduce the broad STDP curve presented in **Mishra et al., 2016**, $\tau_\pm = 62.5$ ms was used. In the classical asymmetric STDP rules, $A_+$ is positive, while $A_-$ is negative; here, both of them were set to 80 pA to obtain a symmetric STDP curve (**Mishra et al., 2016**). In simulations using the asymmetric STDP rule, $\tau_\pm = 20$ ms, $A_+ = 400$ pA, $A_- = -400$ pA, and $w_{max} = 40$ nS were used. In both cases, PCs were sparsely connected (Table 3) and weights were initialized to 0.1 nS. In the learning phase, the intrinsic dynamics of the PCs were not modeled explicitly since only the timing of their spikes mattered, which was set directly as described above. No self-connections were allowed, and diagonal elements of the learned recurrent weight matrix were always set to zero after any modification.

## In vitro electrophysiology

Somatic whole-cell patch-clamp recordings were performed in acute hippocampal slices as described before (**Papp et al., 2013**; **Schlingloff et al., 2014**; **Kohus et al., 2016**). PCs were recorded in the CA3 PC layer of juvenile control mice, while PVBCs were recorded in a targeted manner in transgenic mice that expressed enhanced green fluorescent protein controlled by the parvalbumin promoter (BAC-PV-eGFP) (**Meyer et al., 2002**). To characterize the physiological response properties of the neurons, hyperpolarizing and depolarizing current steps of various amplitudes were injected into the soma, and the voltage response of the cell was recorded. Injected current pulses had a duration of 800 ms, and amplitudes between −100 and 600 pA. Experimental traces were corrected for the theoretical liquid junction potential before further use.

## Single-cell models

Neurons were modeled with the AdExpIF model (**Naud et al., 2008**; **Gerstner et al., 2014**). AdExpIF neurons are described by their membrane potential $V(t)$ and the adaptation variable $w(t)$, which obey

$$C_m \frac{dV(t)}{dt} = -\left( g_L(V(t) - V_{rest}) - g_L \Delta T \exp\left( \frac{V(t) - \vartheta}{\Delta T} \right) + I_{syn}(t) + w(t) \right) \tag{5}$$

$$\tau_w \frac{dw(t)}{dt} = a(V(t) - V_{rest}) - w(t) \tag{6}$$

where $C_m$ is the membrane capacitance, $g_L$ is the leak conductance, $V_{rest}$ is the reversal potential of the linear leak current (which is approximately equal to the resting potential), $\vartheta$ is the intrinsic spike threshold, $\Delta T$ characterizes the 'sharpness' of the threshold, $w(t)$ is the adaptation current, and $I_{syn}$ is the synaptic current (see below). When $V(t)$ crosses the firing threshold $\theta$, it is reset to $V_{reset}$ and remains there for a refractory period $t_{ref}$. The adaptation current is also increased by a factor $b$ at each spike arrival. The parameter $a$ describes the strength of subthreshold adaptation.

To investigate the role of adaptation, an ExpIF PC model was also fit to the data. The ExpIF model is the same as Equation (5) without the $w(t)$ adaptation current (implemented as an AdExpIF model with parameters $a$ and $b$ set identically to zero). The parameters of all models were fit to experimental data from our somatic whole-cell recordings, and the voltage responses to current injections of four different amplitudes (including two subthreshold and two suprathreshold stimuli) were used in each case. Parameters were tuned using the Optimizer package (**Friedrich et al., 2014**) with the NEST simulator as backend (**van Albada et al., 2015**). Spike count, ISI distribution, latency to first spike, and mean squared error (excluding spikes) were used as equally weighted features. After comparing different optimization techniques, the final parameters presented here were obtained with an evolutionary algorithm implemented by the inspyred package (**Garrett, 2012**), running for 100 generations with a population size of 100. The parameters that yield the best models for the CA3 populations are summarized in **Table 2**.

**Table 2.** Optimized parameters of pyramidal cell (PC) (AdExpIF and ExpIF) and parvalbumin-containing basket cell (PVBC) models. Physical dimensions are as follows: $C_m$: pF; $g_L$ and $a$: nS; $V_{rest}$, $\Delta T$, $\vartheta$, $\theta$, and $V_{reset}$: mV; $t_{ref}$ and $\tau_w$: ms; $b$: pA.

| | $C_m$ | $g_L$ | $V_{rest}$ | $\Delta T$ | $\vartheta$ | $\theta$ | $V_{reset}$ | $t_{ref}$ | $\tau_w$ | $a$ | $b$ |
|---|---|---|---|---|---|---|---|---|---|---|---|
| PC | 180.13 | 4.31 | −75.19 | 4.23 | −24.42 | −3.25 | −29.74 | 5.96 | 84.93 | −0.27 | 206.84 |
| PC | 344.18 | 4.88 | −75.19 | 10.78 | −28.77 | 25.13 | −58.82 | 1.07 | - | - | - |
| PVBC | 118.52 | 7.51 | −74.74 | 4.58 | −57.71 | −34.78 | −64.99 | 1.15 | 178.58 | 3.05 | 0.91 |

**Table 3.** Synaptic parameters (taken from the literature or optimized).
Physical dimensions are as follows: $\hat{g}$: nS; $\tau_r$, $\tau_d$, and $t_d$ (synaptic delay): ms; and connection probability $p_{conn}$ is dimensionless. GC stands for the granule cells of the dentate gyrus. ($GC \rightarrow PC$ synapses are referred as mossy fibers.) Sym. and asym. indicate maximal conductance parameters obtained for networks trained with the symmetric and the asymmetric spike-timing-dependent plasticity (STDP) kernel, respectively. PC: pyramidal cell; PVBC: parvalbumin-containing basket cell.

| | $\hat{g}$ | | $\tau_r$ | $\tau_d$ | $t_d$ | $p_{conn}$ |
| --- | --- | --- | --- | --- | --- | --- |
| | Sym. | Asym. | | | | |
| PC$\rightarrow$PC | 0.1–6.3 | 0–15 | 1.3 | 9.5 | 2.2 | 0.1 |
| PC$\rightarrow$PVBC | 0.85 | | 1 | 4.1 | 0.9 | 0.1 |
| PVBC$\rightarrow$PC | 0.65 | | 0.3 | 3.3 | 1.1 | 0.25 |
| PVBC$\rightarrow$PVBC | 5 | | 0.25 | 1.2 | 0.6 | 0.25 |
| GC$\rightarrow$PC | 19.15 | 21.5 | 0.65 | 5.4 | - | - |

## Synapse models

Synapses were modeled as conductances with biexponential kinetics:

$$g(t) = \hat{g}A\left(\exp(-\tfrac{t}{\tau_d}) - \exp(-\tfrac{t}{\tau_r})\right) \tag{7}$$

where $\hat{g}$ is the peak conductance (which will also be referred to as the synaptic weight) and $\tau_r$ and $\tau_d$ are rise and decay time constants, respectively. The normalization constant $A = exp(-\tfrac{t_p}{\tau_d}) - exp(-\tfrac{t_p}{\tau_r})$ was chosen such that the synapses reach their peak conductance at $t_p = \tau_d\tau_r/(\tau_d - \tau_r)log(\tau_d/\tau_r)$ ms. Kinetic parameters were taken from the literature (*Geiger et al., 1997*; *Bartos et al., 2002*; *Lee et al., 2014*; *Vyleta et al., 2016*; *Guzman et al., 2016*) and are summarized in *Table 3*. The postsynaptic current contained AMPA receptor- and GABA-A receptor-mediated components, and was computed as

$$I_{syn}(t) = g_{AMPA}(t)(V(t) - E_{exc}) + g_{GABA}(t)(V(t) - E_{inh}) \tag{8}$$

where $E_{exc} = 0$ mV and $E_{inh} = -70$ mV are the reversal potentials of excitatory and inhibitory currents, respectively.

## Network optimization

Synaptic weights of the network (five parameters in total) were optimized with an evolutionary algorithm using a custom written evaluator in BluePyOpt (*Van Geit et al., 2016*). The multiobjective fitness function $\mathcal{F}$, designed specifically for this network, included six separately weighted features (Equation (9)): physiological PC firing rate, no significant gamma oscillation in the PVBC population, significant ripple frequency oscillations in the rates of PC and PVBC populations, as well as high ripple vs. gamma power in the rates of the PC and PVBC populations:

$$\mathcal{F} = \left[ \quad \exp\left(-\frac{(\nu_{PC} - 2)^2}{2 \times 0.5^2}\right), \quad \delta(f_{\gamma PVBC}), \quad \exp\left(-\frac{(f_{rPC} - 180)^2}{2 \times 20^2}\right), \right.$$
$$\left. 2 \times \exp\left(-\frac{(f_{rPVBC} - 180)^2}{2 \times 20^2}\right), \quad \frac{\sum P(\omega_{rPC})}{\sum P(\omega_{\gamma PC})}\Big|_{[0,5]}, \quad 2 \times \frac{\sum P(\omega_{rPVBC})}{\sum P(\omega_{\gamma PVBC})}\Big|_{[0,5]}\right] \tag{9}$$

where $\nu$ is the firing rate; $\delta(f_{\gamma PVBC})$ is 1 if no significant gamma oscillation was detected in the PVBC population rate, and 0 otherwise; $f_r$ and $f_\gamma$ are significant peaks in the ripple and gamma range (see below) of the PSD of the firing rate, respectively; $P(\omega_r)$ and $P(\omega_\gamma)$ are the periodogram values within the gamma and ripple bands of the firing rate, respectively, while the sums represent the total power within the relevant frequency bands (as below). Ripple/gamma power ratios used for the optimization were clipped to fall into [0, 5]. As in the case of the network simulations (see above), spectral features were extracted only in those time windows when the PC firing rate exceeded the 2 Hz high-activity state detection threshold. Sequence replay was not analyzed during the optimization. Optimizations were run with 50 offspring for 10 generations. Synaptic weights were varied in the [0.1, 5] nS range,

except the 'detonator' mossy fiber ones that were given a higher [15, 30] nS range (**Henze et al., 2002**; **Vyleta et al., 2016**). For the learned recurrent excitatory weights, an additional scaling factor was introduced. All learned weights are presented with this optimized scale factor (0.62 for symmetric and 1.27 for asymmetric STDP rule) taken into account. Final weights are presented in **Table 3**. The ExpIF PC model required much higher synaptic drive to fire at the same frequency as the AdExpIF model, thus the mossy fiber input weight was doubled (38.3 nS) when ExpIF PC models were used.

## LFP estimate

An estimate of the LFP was calculated by summing the synaptic currents of a randomly selected subset of $N = 400$ PCs (**Mazzoni et al., 2008**). This approach is essentially equivalent to using 'transmembrane' currents to calculate the field potential at an arbitrary sampling point $x_e$ using volume conduction theory and the forward model (**Einevoll et al., 2013**):

$$V(x_e, t) = \frac{1}{4\pi\sigma} \sum_{n=1}^{N} \frac{I_n(t)}{|x_e - x_n|} \tag{10}$$

where $\sigma = 1/3.54$ S/m is the extracellular conductivity and $I_n(t)$ denotes the transmembrane currents of the $n$th neuron. There was no attempt to replicate the spatial organization of CA3 PCs, and a uniform $|x_e - x_n| = 1 \mu m$ distance from the sampling point was used (note that this choice affects the results only as a constant scaling factor). The resulting signal was low-pass filtered at 500 Hz with a third-order Butterworth filter.

## Spectral analysis

Power spectral density (PSD) was estimated by Welch's method with a Hann window using 512 long segments in case of population rates (sampling frequency = 1 kHz) and 4096 long segments for LFP (see below, sampling frequency = 10 kHz) with 0.5 overlap. If the network showed multiple sequence replays during the 10-s-long simulations (most cases), only the detected high-activity states (see above) were analyzed and averaged to get rid of the high power at ~1.5 Hz, signaling the frequency of sequence replays. In this case, shorter segments (256 and 2048, respectively) were used to estimate the PSD. The significance of peaks in the power spectra in the gamma (30–100 Hz) and ripple (150–220 Hz) bands was evaluated using Fisher's g-statistic (**Fisher, 1929**) defined as

$$g = \frac{\max_k(P(\omega_k))}{\sum_{k=1}^{N} P(\omega_k)} \tag{11}$$

where $P(\omega)$ is the periodogram (Welch's method estimates PSD by averaging periodograms from the short segments) evaluated at $k$ discrete frequencies, $N$ is the length of the periodogram, and $\sum_{k=1}^{N} P(\omega_k)$ is the total power of the spectrum. The distribution of g-statistics under the null hypothesis ($H_0$) (Gaussian white noise) is given by

$$p = Pr(g^*) = \sum_{k=1}^{b} (-1)^{k-1} \frac{N!}{k!(N-k)!} (1 - kg)^{N-1} \tag{12}$$

where $b$ is the largest integer less than $1/g$. Large value of $g$ (small p-value) indicates a strong periodic component and leads to the rejection of $H_0$. Alpha level 0.05 was used throughout the study. To characterize nonsignificant oscillations too, gamma and ripple power (defined as the sum in the given frequency band divided by the total power in the 0–500 Hz range) were calculated as well. Time-frequency representations were created by convolving the firing rates (or LFP) with samples of the integral of the Morlet wavelet $\Psi(t) = exp(-t^2/2)cos(5t)$ evaluated at the scales corresponding to the 25–325 Hz band using the pywt package (**Lee et al., 2006**).

## Replay analysis

Sequence replay was analyzed with methods used by experimentalists having access to spike times of hundreds of identified neurons (**Olafsdottir et al., 2018**). Firstly, candidate replay events were selected based on the averaged (into 20 ms bins) PC population firing rate crossing the threshold of 2 Hz for at least 260 ms. Secondly, the animal's position was estimated with a memoryless Bayesian place decoder based on the observed spikes within the selected time windows (**Davidson et al., 2009**; **Karlsson and Frank, 2009**). Only spikes from the $N = 4000$ place cells were used. For numerical stability, log likelihoods were calculated:

$$\log(Pr(spikes|x)) = \sum_{i=1}^{N} n_i \log\left(\frac{\Delta T \tau_i(x)}{n_i!}\right) - \Delta T \sum_{i=1}^{N} \tau_i(x) \tag{13}$$

where $n_i$ is the number of spikes of the th neuron within the $\Delta T = 10$-ms-long, nonoverlapping time bins, and $\tau_i(x)$ is the tuning curve used for spike train generation (Equation (1)). The 3-m-long linear track was binned into 50 intervals, resulting in 6 cm spatial resolution. Thirdly, constant velocity $v$ neural trajectories were detected with a 2D band finding method in the decoded posterior matrix (*Davidson et al., 2009*). For candidate events consisting of $n$ time bins, the average likelihood $R$ that the animal is within distance $d = 18$ cm of a particular trajectory is given by

$$R(v, x_0) = \frac{1}{n} \sum_{k=0}^{n-1} Pr(|x - (x_0 + v_k \Delta T)| \leq d) \tag{14}$$

where $x_0$ is the starting position of the trajectory. $R(v, x_0)$ was maximized using an exhaustive search to test all combinations of $v$ between –18 m/s and 18 m/s in 0.3 ms/s increments (excluding slow trajectories with speed $\in [-0.3, 0.3]$ m/s) and $x_0$ between –1.5 m and 4.5 m in 3 cm increments. Lastly, to determine the significance of replay, $R_{max}$ was compared to the best-line fits of 100 posterior probability matrices generated by shuffling the identities of cells included in the candidate event. Only events with $R_{max}$ values exceeding the 95th percentile of their own shuffled distribution were labeled as replay.

To replicate the step-size analysis of *Pfeiffer and Foster, 2015*, the position of the animal was estimated as a weighted average based on the posterior matrix in each time bin instead of the band finding method. As their control distribution for the skewed step sizes ('predicted step-size' distribution) was derived for a 2D arena, it was not directly applicable to our linear track setup. Therefore, we defined the predicted step-size distribution based on the ratio of the length of the replayed path and the duration of the replay event for the SWRs detected in the simulations.

## Accessibility

The source code is publicly available at https://github.com/KaliLab/ca3net (copy archived at ,swh:1:rev:bfeb4fa6c8d07f75778df671cc9500cf0316df51, *Ecker, 2022*).

## Acknowledgements

We thank G Buzsáki, D Csordás, G Nyíri, B Ujfalussy, and V Varga for insightful comments on the manuscript, M Stimberg for his help with Brian2, and F Crameri for colormaps.

## Additional information

### Funding

| Funder | Grant reference number | Author |
| --- | --- | --- |
| Hungarian Scientific Research Fund | K83251 | Attila I Gulyás<br>Szabolcs Káli<br>Mária Karlócai |
| Hungarian Scientific Research Fund | K85659 | Orsolya I Papp<br>Norbert Hájos |
| Hungarian Scientific Research Fund | K115441 | Attila I Gulyás<br>Szabolcs Káli |
| Hungarian Medical Research Council | 2017-1.2.1-NKP-2017-00002 | Norbert Hájos |
| European Commission | ERC 2011 ADG 294313 | Tamás Freund<br>Attila I Gulyás<br>Szabolcs Káli |
| European Commission | 604102 | Tamás Freund<br>Attila I Gulyás<br>Szabolcs Káli |

| Funder | Grant reference number | Author |
|---|---|---|
| European Commission | 720270 | Tamás F Freund<br>Attila I Gulyás<br>Szabolcs Káli |
| European Commission | 785907 | Tamás F Freund<br>Attila I Gulyás<br>Szabolcs Káli |
| National Research, Development and Innovation Office | Artificial Intelligence National Laboratory | Szabolcs Káli |

The funders had no role in study design, data collection and interpretation, or the decision to submit the work for publication.

## Author contributions

András Ecker, Conceptualization, Formal analysis, Investigation, Methodology, Software, Visualization, Writing – original draft, Writing – review and editing; Bence Bagi, Data curation, Formal analysis, Investigation, Methodology, Software, Writing – review and editing; Eszter Vértes, Conceptualization, Formal analysis, Investigation, Methodology, Software, Writing – review and editing; Orsolya Steinbach-Németh, Formal analysis, Investigation, Writing – review and editing; Mária R Karlócai, Investigation, Writing – review and editing; Orsolya I Papp, Data curation, Investigation, Writing – review and editing; István Miklós, Formal analysis, Methodology, Writing – review and editing; Norbert Hájos, Attila I Gulyás, Conceptualization, Data curation, Writing – review and editing; Tamás F Freund, Conceptualization, Funding acquisition, Writing – review and editing; Szabolcs Káli, Conceptualization, Data curation, Funding acquisition, Investigation, Methodology, Supervision, Writing – original draft, Writing – review and editing

## Author ORCIDs

András Ecker (iD) http://orcid.org/0000-0001-9635-4169
Szabolcs Káli (iD) http://orcid.org/0000-0002-2740-6057

## Ethics

Experiments were approved by the Committee for the Scientific Ethics of Animal Research (22.1/4027/003/2009) and were performed according to the guidelines of the institutional ethical code and the Hungarian Act of Animal Care and Experimentation. Experiments were performed in acute brain slices; no animal suffering was involved as mice were deeply anaesthetized with isoflurane and decapitated before slice preparation. Data recorded in the context of other studies were used for model fitting, and therefore no additional animals were used for the purpose of this study.

## Decision letter and Author response

Decision letter https://doi.org/10.7554/eLife.71850.sa1
Author response https://doi.org/10.7554/eLife.71850.sa2

# Additional files

## Supplementary files

• Transparent reporting form

## Data availability

The source code to build, run and analyze our model is publicly available on GitHub: https://github.com/KaliLab/ca3net, (copy archived at swh:1:rev:bfeb4fa6c8d07f75778df671cc9500cf0316df51).

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
