## [Editor Report]

This manuscript will be of interest to theoretical neuroscientists broadly defined and potentially also to experimentalists investigating the hippocampus. A biologically plausible spiking neural network model of subregion CA3 of the hippocampus is proposed and studied in order to pinpoint the mechanistic sources of sharp wave ripples and replay observed in vivo.

---

## [Decision Letter]

**Decision letter after peer review:**

Thank you for submitting your article "Hippocampal sharp wave-ripples and the associated sequence replay emerge from structured synaptic interactions in a network model of area CA3" for consideration by *eLife*. Your article has been reviewed by 3 peer reviewers, and the evaluation has been overseen by a Reviewing Editor and Laura Colgin as the Senior Editor. The following individuals involved in review of your submission have agreed to reveal their identity: Daniel McNamee (Reviewer #1); Bruce Graham (Reviewer #3).

Essential revisions:

The reviewers agree that this manuscript is worthy of acceptance, but that the authors should be encouraged to further clarify or explain the mechanisms causing ripples and replay, as detailed in their responses below. The main broad issues for consideration are: (1) What causes forward versus backwards replay; (2) The necessity or otherwise for the recurrent connectivity created by the specific STDP rule used in generating replay; (3) The necessity of adaptation in causing replay, when cells might fire only once per event; (4) Whether the contribution of an extra inhibitory neuronal cell type might produce a more accurate model of the duration of SWRs; (5) Whether any explicit experimental predictions can be made.

*Reviewer #1 (Recommendations for the authors):*

Thank you for the well-written manuscript and also for providing your source code. I had a few queries and suggestions that I hope may be helpful.

1. What is the functional role of the non-spatial PCs?

2. It seems that the heavy-tailed step size distribution of Pfeiffer and Foster is replicated (line 160). Can the gaussian step size distribution of diffusive reactivations be induced? (observed during rest, see Stella et al., Neuron, 2019). Furthermore, beyond the bidrectionality of replay, recent experiments have demonstrated that sequential hippocampal reactivations make take on even more exotic forms such as generative cycling (Kay et al., Cell, 2020). It would be interesting to speculate how such flexible dynamism may emerge in this circuit model e.g. based on entorhinal input.

3: it seems (line 538) that the exploring rodents moved at a constant velocity. If this interpretation is correct, Im wondering if the replay phenomenology is robust to heterogeneous velocities during naturalistic movement (e.g. stop/starts)?

4: Can the model be successfully deployed in an open box environment?

*Reviewer #2 (Recommendations for the authors):*

Below is a list of questions and comments in ascending order with respect to the line number. I have marked some with an 'S' to highlight that I find them pretty significant. Some of these points are already made in the Public Review and I allow myself to repeat them here so that they are easier to understand. Apologies if this unduly increases the length of the review.

l. 95- 96: Could the authors highlight the main characteristics of the connection probabilites in their model? What are their typical numerical values? They are given in Methods, but I think it is nicer to have an idea about them without having to go consult that section.

(S) Figure 1: How important is it that the STDP kernel is temporally broad (broader than the timescale of a ripple oscillation)?

l. 110: To which figures does the reference refer to? To Figure 1A or its Supplement? Or is Methods and Figure 1A simultaneously referenced? Then, 'Methods and Figure 1A' would be more appropriate. Please clarify.

l. 130: Chenkov et al., also use a symmetric learning rule (see their Figure 8) supporting reverse replay, so this part of the manuscript is not novel. The authors should rephrase this part to clarify the point.

After l. 134: it should be made clearer what the 'spatially and temporally unstructured' external inputs to the network are, and which cells (all? only a fraction?) are receiving them.

l. 172: At least in CA1, PVBCs fire at the ripple frequency of 200 Hz, see e.g. Ylinen et al., 1995. Is this comparatively low inhibitory population firing rate important for the model?

l. 178: Are the weights scaled after the initial learning phase, or before? Please clarify this.

l. 192: The reference should be to Supplementary Figure 3 since there are no spectral measures shown in Supplementary Figure 1.

(S) l. 195: The main result of Stark et al., 2014 is that interactions between PCs and PVBCs underlie ripple oscillations, and that both cell types are needed for ripples. So citing this reference when saying that ripples start from the PVBC population and then propagate to the pyramidal cells does not seem to be right.

Figure 3B bottom plot: To which scaling factor does 100% LFP power correspond to?

l. 222: Did Theodoni et al., show that one needs a symmetric STDP kernel to generate both forward and reverse replay? If so, it would be good to state this explicitly.

l. 238 ff. Is it important that a 97-3 split is done on the weight matrix? What would happen if, for example, a 50-50 split was done? The choice for the split seems arbitrary and should be better justified.

(S) l. 251: This reads as if the PC-PC coupling structure 'does everything', i.e. it generates replay but also ripple oscillations. This raises the interesting question whether ripples could in principle also be observed in unstructured networks. Could the authors comment on this (for example, by showing a PC spike raster for Figure 6F, where ripple oscillations seem to be present for large PC weight multipliers)?

(S) l. 258 ff. There seems to be a dichotomy revealed by the model: the PC-PC connections are crucial, but the ripple oscillation is generated by PVBCs and is only present in the LFP, but not in the PC firing activity- although the model was designed to show ripple oscillations in the PC activity (Equation 9). Is there an easy intuitive reason for why this is the case?

Figure 7B: It is confusing that the PCs with the highest indices in the rasterplot have the same color (orange) as the BCs in the scheme on top of the figure. Similarly at the top of Figure 2.

(S) l. 295: Similarly to comment about l. 195, Stark et al., 2014 seem to be cited incorrectly: in that study, driving PVBCs did not result in ripple frequency oscillations in the LFP, see their Figure 5. E and F and 6.

l. 321-322: This reads like an addition, while it is a summary of the paragraph above. Please consider re-writing to reflect this.

(S) l. 262: The spike-triggered adaptation must be quite strong because a single PC spikes only once or twice during a given ripple, so the adaptation variable gets only one kick and then decays. This means that either the adaptation starts already at quite a high value or the kick size is large, because otherwise there would be no effect on the membrane voltage. Could the authors comment on this?

l. 347- 348: What is a stochastically initiated buildup period? Was it described before?

l. 365: Given that after learning, the distribution of the PC-PC synaptic strengths is also long-tailed, it might well be that the effective connectivity is lower than 10% because only strongly connected cells fire and the rest stays silent. Is this plausible?

l. 422: Maybe the authors would find the study 'Generation of Sharp Wave-Ripple Events by Disinhibition' (Evangelista et al., J Neurosci 2020) interesting. This study contains a mechanism for the generation of SPW/R events by disinhibition of PCs by a third inhibitory cell population. Here, short-term depression acts on the connection between PVBCs and the third cell population.

Table 3: The decay time constant for PC-PC synapses is large (more than 5 ms). Is this a realistic value for AMPAergic synapses?

(S) l. 604/ 620 (and also the section beginning on l. 291): The network was optimized to generate strong ripple-frequency oscillations in both the PC and PVBC population (Equation 9). In Figure 1D, however, it looks as if there was no ripple oscillation in the PC rate, but only in the LFP. Could the authors please clarify this? It would be nice to see a magnification of the spike rasterplot at the top of Figure 1D. I also wonder whether there could be a ripple oscillation present in the spiking activity of PCs shown in Figure 8C. It isn't present in the LFP as shown in Figure 8D, but since the network was optimized to display ripple oscillations in the PC activity (Equation 9) and since there still is bi-directional replay in the model, it might well be that there are still ripple oscillations in the PC population activity, as shown in Figure 3—figure supplement 1 for a different setup. It is also not defined how the population rates for PCs and PVBCs are calculated, but the temporal resolution (for example in Figure 1D middle plot) looks rather low and this might mask fast oscillations. Could the authors comment on this?

Figure 2, supplement 2: it looks like some fraction of PC and PVBCs is bursting (very small ISIs). Are they important for the network dynamics? Is the adaptation maybe only effective in the bursting PCs? This might be important to gain a better mechanistic understanding of the model.

*Reviewer #3 (Recommendations for the authors):*

The authors are careful to consider some of the limitations of their simple model. However, their study would certainly be strengthened if some of these and other questions could be addressed. Specific consideration should be given to the following:

Spike adaptation alone does not result in a match to the experimentally-recorded durations of SWRs, as the authors acknowledge. It would be good at least to see results from the simulations that have been tried using an extra inhibitory neuron type to try to address this mismatch; and other mechanisms also could be explored here.

What specific spatio-temporal characteristics of spike activity trigger a sequence replay? For example, how sensitive is this to the number of closely associated place cells that spike contemporaneously and does it require some sequencing in this spiking? Is it such initial sequencing that determines whether forward or backward replay occurs?

As discussed, there are biases towards forward or backward replay in different animal situations. How can such biases arise in this model? The "cued" replay results (Figure 2C) demonstrate the obvious of forward replay if the cue is at the track beginning, with backward replay if the cue is at the end of the track, but how in general may such "beginnings" and "ends" be defined in terms of place cells? The strict beginning and end here is somewhat artificial. How can a bias for forwards or backwards replay be introduced at intermediate points along the track? Insights into the characteristics of spike activity triggering replay, as asked for above, should help answer this.

Why do we not see simultaneous forward and backward replay from a single trigger point?

---

## [Author Response]

Essential revisions:The reviewers agree that this manuscript is worthy of acceptance, but that the authors should be encouraged to further clarify or explain the mechanisms causing ripples and replay, as detailed in their responses below. The main broad issues for consideration are: (1) What causes forward versus backwards replay;

This question was explored explicitly in the simulations shown in Figure 2C, which demonstrated that one can evoke either forward or backward replay via external input that corresponds to neural activity at the start and the end of the track, respectively. Without any cuing input we find (as expected from the symmetry of the plasticity rule and all other properties of the network) that forward and backward replays occur with the same frequency. At a more general level, the direction of replay in the model can depend on systematic biases and random variations in the weight structure, but also on (random or deterministic) spatio-temporal structure in the external inputs. A more detailed discussion of this issue has been incorporated into the Discussion section of the paper, and also appears in our responses to the reviewers below.

2) The necessity or otherwise for the recurrent connectivity created by the specific STDP rule used in generating replay;

A key finding of our work is that sufficiently strong, properly structured recurrent excitatory connections are essential for several key aspects of network dynamics, including sequence replay, sharp waves, and (indirectly, by causing a high level of PVBC activity) ripple oscillations. In principle, such structured connectivity could be formed by a variety of different mechanisms (as discussed in the section “Biological plausibility of the model”). The plasticity rule that we used in most of our simulations is based on direct measurement of STDP in pairs of CA3 pyramidal neurons in hippocampal slices (Mishra et al., Nature Comms., 2016). To our best knowledge this is the only rule that has been experimentally measured in area CA3. On the other hand, we do investigate the effect of using a different type of STDP rule (one that has been measured in several other types of synapse), and find that this choice leads to altered dynamics and the absence of reverse replay events (Figure 5). Therefore, our results confirm that applying the experimentally determined plasticity rule to activity patterns seen during exploration leads to a pattern of synaptic interactions that gives rise to the correct form of network dynamics (including sharp wave-ripples and bidirectional replay) in off-line states of the hippocampus.

3) The necessity of adaptation in causing replay, when cells might fire only once per event;

In our model, adaptation is necessary for sequence replay as it destabilizes a stationary bump of activity, and causes it to move on to previously inactive populations with nearby place fields. This is clearly demonstrated by Figure 7, which shows that the activity bump remains stationary without adaptation in PCs. On the other hand, spike-triggered adaptation does not need to be dominant in single cells to shift network activity away from the currently active population, and we have confirmed this by looking at synaptic and adaptation currents in some randomly selected neurons. Excitation and inhibition are almost balanced in active cells, and the addition of the adaptation current is sufficient to silence them. By contrast, previously inactive cells with nearby place fields still get strong excitatory input, so that they can start to fire if they were not recently active (and thus have no adaptation current). In addition, the majority of place cells in our model fire more than once during individual sharp waves, although we do not believe this to be essential for the main results of the paper. We have amended the manuscript to clarify this fact, and also added an insert to panel B1 and a new panel C to Supplementary Figure 4.

4) Whether the contribution of an extra inhibitory neuronal cell type might produce a more accurate model of the duration of SWRs;

The mechanisms responsible for the termination of SWRs are currently unknown, and this is also an open question for our current model. Several different mechanisms based on cellular adaptation, short-term synaptic plasticity, and multiple inhibitory cell populations have been proposed. An additional possible mechanism based on the (learned or hard-wired) structure of recurrent excitation is demonstrated in this paper. As we briefly mentioned in the previous version of the manuscript, we also found that adding a second inhibitory population that provides delayed, long-lasting negative feedback to the PCs could explain the termination of SWRs. However, these simulations were performed in an earlier version of the circuit model that lacked learned excitatory connections, and we found that introducing this mechanism in the final model required substantial retuning of several parameters. Therefore, we decided to defer the full exploration of possible SWR termination mechanisms to a subsequent study; we have removed the reference in the manuscript to the preliminary simulation results, and cite related published work (Evangelista et al., J. Neurosci. 2020) to back the proposed idea.

5) Whether any explicit experimental predictions can be made.

Both our concrete mechanistic model and the general conceptual framework suggested by our simulation results can be used to derive novel experimental predictions. In fact, the consequences of all the manipulations that we employed to probe our *in silico* network can be regarded as experimental predictions. Although some of these manipulations (such as shuffling the weights) would be difficult to replicate in real experiments, others are probably feasible using modern tools (cell-type-specific targeting, optogenetics, chemogenetics, etc.). We have listed several specific predictions of our modeling work in a new section of the Discussion (“Experimental predictions”). These predictions include the following:

– Blocking PVBC-PVBC synaptic interactions should eliminate ripples but not sharp waves or replay, while blocking recurrent excitation is expected to eliminate sharp waves, ripple oscillations and spontaneous replay as well.

– The nature of replay (e.g., stereotypical vs. diffusive) should depend on the task and behavior during learning (e.g., 1D vs. 2D environment, goal-oriented vs. random foraging task).

– It should be possible to influence the content of replay by providing structured input either during learning or before/during SWRs.

– Any manipulation changing the symmetry of the STDP rule should bias the direction of replay.

Reviewer #1 (Recommendations for the authors):Thank you for the well-written manuscript and also for providing your source code. I had a few queries and suggestions that I hope may be helpful.1. What is the functional role of the non-spatial PCs?

When we constructed the model, we included PCs that do not have place fields in a particular environment to better match the relevant experimental data. This also allowed us to check whether these cells have any functional role in our model; however, we found that (at least in the current, simplified model where cells without place fields fired at low rates during exploration) non-spatial neurons had only weak excitatory interactions with other PCs, and were not significantly involved in the generation of SWRs or replay. On the other hand, when we investigated the effects of learning in multiple environments, place cells were selected independently and randomly; in this case, the presence of non-spatial cells decreased the amount of overlap between representations of the two environments, which may have contributed to the robustness of independent replay of spatial sequences in the two environments (although we have not tested this explicitly).

2. It seems that the heavy-tailed step size distribution of Pfeiffer and Foster is replicated (line 160). Can the gaussian step size distribution of diffusive reactivations be induced? (observed during rest, see Stella et al., Neuron, 2019). Furthermore, beyond the bidrectionality of replay, recent experiments have demonstrated that sequential hippocampal reactivations make take on even more exotic forms such as generative cycling (Kay et al., Cell, 2020). It would be interesting to speculate how such flexible dynamism may emerge in this circuit model e.g. based on entorhinal input.

Thank you for drawing our attention to this interesting issue. In our model, the nature of replayed sequences depends mostly on the activity patterns of hippocampal neurons during exploration (which determine the pattern of recurrent interactions), but it is also influenced by (either sensory-driven or random) input before/during reactivation episodes. In the basic scenario explored in our paper, learning occurred during repeated runs along a single linear track, which led to reactivations consisting of random chunks of the single learned trajectory, and allowed our model to replicate the replay dynamics observed by Pfeiffer and Foster. However, other types of learning experience would be expected to result in different reactivation dynamics. For instance, the random 2D exploration investigated by Stella et al., Neuron, (2019) may be expected to result in a 2D bump attractor structure encoded in the recurrent weights, and this, combined with adaptation and some random noise, may explain the diffusive nature of reactivation in their experiments. Similarly, the learning of alternative routes in the experiments of Kay et al. Cell., (2020), combined with adaptation, may explain the spontaneous switching between representations of alternative future paths. We have included these thoughts in the Discussion (lines 384-403).

3: It seems (line 538) that the exploring rodents moved at a constant velocity. If this interpretation is correct, Im wondering if the replay phenomenology is robust to heterogeneous velocities during naturalistic movement (e.g. stop/starts)?

At an intuitive level, if velocity varies more or less randomly across trials, and the starts and stops occur at different locations in each trial, then replay should be robust given enough repetitions of the same path (the current results are based on more than 40 repetitions), although we have not tested this explicitly. Robustness against variations in speed is also increased by the broad temporal kernel of the STDP rule. On the other hand, if starts and stops occurred at stereotypical locations, our model predicts that these locations would become preferred starting points (or endpoints) of replayed paths. In fact, it would be relatively straightforward to modify the spike patterns used during the exploration phase to model variations in speed.

4: Can the model be successfully deployed in an open box environment?

In principle, exploration in an open box (or any other environment) could be modeled by changing the spike statistics during the learning phase so that it mimics the corresponding experimental observations. All other properties and parameters of the model would remain unchanged. This would also allow us to test whether we can predict the replay dynamics in these novel situations (see above), and we intend to examine this in a follow-up project.

Reviewer #2 (Recommendations for the authors):Below is a list of questions and comments in ascending order with respect to the line number. I have marked some with an 'S' to highlight that I find them pretty significant. Some of these points are already made in the Public Review and I allow myself to repeat them here so that they are easier to understand. Apologies if this unduly increases the length of the review.l. 95- 96: Could the authors highlight the main characteristics of the connection probabilites in their model? What are their typical numerical values? They are given in Methods, but I think it is nicer to have an idea about them without having to go consult that section.

We decided to list most parameter values in the Methods rather than the Results section so that the Results can remain focused on the main ideas and the new results. More specifically, we do not think that the precise connection probabilities have a major qualitative effect on the behavior of the model, and their biological values are also to some extent uncertain, so we prefer to keep them in the Methods, but added an internal reference (line 97) to the manuscript to facilitate the localization of this information.

(S) Figure 1: How important is it that the STDP kernel is temporally broad (broader than the timescale of a ripple oscillation)?

The exact width of the STDP kernel is probably not critical in our model (its symmetry is more important for our results) although it is likely to influence the exact replay dynamics (e.g., the speed of activity propagation). However, we did not investigate this issue in detail here because the effects of the shape of the STDP kernel of network dynamics were recently analyzed by Theodoni et al., *eLife*, (2017).

l. 110: To which figures does the reference refer to? To Figure 1A or its Supplement? Or is Methods and Figure 1A simultaneously referenced? Then, 'Methods and Figure 1A' would be more appropriate. Please clarify.

We thank the reviewer for drawing our attention to this issue; we have added “and” to this reference in the manuscript (and to all similar cases as well).

l. 130: Chenkov et al., also use a symmetric learning rule (see their Figure 8) supporting reverse replay, so this part of the manuscript is not novel. The authors should rephrase this part to clarify the point.

We first note that Chenkov et al., PLOS, (2017) did not use a learning rule to set up the excitatory recurrent connections. The authors constructed hard-wired chains of assemblies and varied the connection probabilities within and between assemblies (and used inhibitory plasticity to balance the network) As a last experiment they showed that symmetric connections can lead to reverse replay, but only if it is cued (as shown in their Figure 8). We have clarified this point in the manuscript (lines 291-293). Besides, we do not claim to be the first to observe reverse replay with a symmetric learning rule (see e.g., the citation of Haga et al., *eLife*, (2018) in lines 565-568).

After l. 134: it should be made clearer what the 'spatially and temporally unstructured' external inputs to the network are, and which cells (all? only a fraction?) are receiving them.

These inputs represent the spontaneous (random) activation of all synapses onto all CA3 PCs, although, for simplicity, we use only a single type of synapse in the model whose parameters best match those of mossy fiber inputs. We have added a reference to the Methods (where the nature of external inputs is discussed in detail) in line 139 to clarify this point.

l. 172: At least in CA1, PVBCs fire at the ripple frequency of 200 Hz, see e.g. Ylinen et al., 1995. Is this comparatively low inhibitory population firing rate important for the model?

According to more recent publications that recorded the firing activity of identified PVBCs during SWRs, their firing rate is typically substantially less than 200 Hz; e.g., Varga et al., PNAS, (2012) measured 75 Hz in awake mice, and Hájos et al., J. Neurosci., (2013) measured 82 Hz in the CA3 region of the mouse hippocampal slice. These values are in good agreement with our simulation results. Moreover, the exact frequency of PVBC firing is not important in our model; the PVBCs increase their rate, and skip fewer and fewer ripple cycles, as their input is increased.

l. 178: Are the weights scaled after the initial learning phase, or before? Please clarify this.

They are scaled after learning; please see lines 196-197.

l. 192: The reference should be to Supplementary Figure 3 since there are no spectral measures shown in Supplementary Figure 1.

We thank the Reviewer for catching this mistake, resulting from an error in the LaTeX source file. The error has been corrected.

(S) l. 195: The main result of Stark et al., 2014 is that interactions between PCs and PVBCs underlie ripple oscillations, and that both cell types are needed for ripples. So citing this reference when saying that ripples start from the PVBC population and then propagate to the pyramidal cells does not seem to be right.

Stark et al., Neuron, (2014) show that activation of PVBCs induces ripple-frequency coherent spiking of PVBCs and PCs, although this is not detectable in the LFP. They say: “although tonic light activation of PV interneurons cannot induce LFP ripples, it can organize neuronal ensemble spiking into coherent ripple-frequency oscillations, consistent with a modified version of an INT-INT-based timing mechanism (Figure 1D).”

Figure 3B bottom plot: To which scaling factor does 100% LFP power correspond to?

None – this plot shows the percentage of power in the ripple range relative to the total power of the signal for each multiplier. 100% power is the total power within the whole 0-500 Hz range as explained in the Methods.

l. 222: Did Theodoni et al., show that one needs a symmetric STDP kernel to generate both forward and reverse replay? If so, it would be good to state this explicitly.

No, they did not. We have clarified this in line 239.

l. 238 ff. Is it important that a 97-3 split is done on the weight matrix? What would happen if, for example, a 50-50 split was done? The choice for the split seems arbitrary and should be better justified.

The split is indeed arbitrary. We have tried several splits in the 99-1 and 90-10% range; the results were qualitatively similar, but this one gave the cleanest result. There is definitely a tradeoff between the number (more the better) and strength (the fewer averaged the better) of the near-diagonal synapses, but we did not think that the results were interesting enough to be presented in the article. The important point here is that this is a manipulation of the weight matrix that preserves the global structure but completely changes the weight statistics.

(S) l. 251: This reads as if the PC-PC coupling structure 'does everything', i.e. it generates replay but also ripple oscillations. This raises the interesting question whether ripples could in principle also be observed in unstructured networks. Could the authors comment on this (for example, by showing a PC spike raster for Figure 6F, where ripple oscillations seem to be present for large PC weight multipliers)?

In our baseline model, ripples are indeed strongly tied to replay because PVBCs receive a sufficiently large excitatory drive only during these events (corresponding to sharp waves). Ripples can also be observed in unstructured networks (which lack replay), but only with much (3.5 times) stronger weights. For some additional discussion of this issue, please see our response to point 1 in the Public Review section above.

(S) l. 258 ff. There seems to be a dichotomy revealed by the model: the PC-PC connections are crucial, but the ripple oscillation is generated by PVBCs and is only present in the LFP, but not in the PC firing activity- although the model was designed to show ripple oscillations in the PC activity (Equation 9). Is there an easy intuitive reason for why this is the case?

As we discussed in our response to question 3 in the Public Review section, the second part of the statement above is incorrect (ripples are indeed present in the PC rate as shown in Supplementary Figure 5), which solves the apparent paradox.

Figure 7B: It is confusing that the PCs with the highest indices in the rasterplot have the same color (orange) as the BCs in the scheme on top of the figure. Similarly at the top of Figure 2.

We thank the reviewer for pointing out this issue, and we have changed the color of the PVBCs in all figures.

(S) l. 295: Similarly to comment about l. 195, Stark et al., 2014 seem to be cited incorrectly: in that study, driving PVBCs did not result in ripple frequency oscillations in the LFP, see their Figure 5. E and F and 6.

As we also indicated in our response above, Stark et al., Neuron, (2014) do find that PVBC activation induces coherent ripple-frequency spiking activity (although this is not reflected by the LFP, presumably because of insufficient PC activity). As our argument here is about ripple-frequency synchrony and not the LFP, we think that this citation is appropriate.

l. 321-322: This reads like an addition, while it is a summary of the paragraph above. Please consider re-writing to reflect this.

This sentence (saying that recurrent inhibitory connections are both necessary and sufficient for ripple oscillations) is actually meant to be an addition to (or amendment of) the previous sentence (which is only about sufficiency), while it is also a concise summary of the results presented in this section.

(S) l. 262: The spike-triggered adaptation must be quite strong because a single PC spikes only once or twice during a given ripple, so the adaptation variable gets only one kick and then decays. This means that either the adaptation starts already at quite a high value or the kick size is large, because otherwise there would be no effect on the membrane voltage. Could the authors comment on this?

First, we note that spike-triggered adaptation does not need to be dominant in single cells to shift network activity away from the currently active population, and we have confirmed this by looking at synaptic and adaptation currents in some randomly selected neurons. Excitation and inhibition are almost balanced in active cells, and the addition of the adaptation current is sufficient to silence them. By contrast, previously inactive cells with nearby place fields still get strong excitatory input (plus slightly less inhibitory input), so that they can start to fire if they were not recently active (and thus have no adaptation current). In addition, the majority of place cells in our model fire more than once during individual sharp waves. We have amended the manuscript to clarify this fact in lines 188-191 and also added an insert to panel B1 and a new panel C to Supplementary Figure 4.

l. 347- 348: What is a stochastically initiated buildup period? Was it described before?

The stochastic initiation of SWRs was described and discussed extensively by Schlingloff et al., J. Neurosci., (2014). We have now added this citation to this point of the manuscript in line 366.

l. 365: Given that after learning, the distribution of the PC-PC synaptic strengths is also long-tailed, it might well be that the effective connectivity is lower than 10% because only strongly connected cells fire and the rest stays silent. Is this plausible?

We first note that there is no special population of strongly connected place cells in the model, and most place cells participate in replay events (although there are differences due to random variations of spiking activity in the learning phase). On the other hand, it is true that most PCs have a small number of (incoming and outgoing) large weights and many smaller ones, and many of the anatomically existing connections may be too weak to be reliably detected in paired recording experiments. We have added this point to the manuscript in lines 453-455.

l. 422: Maybe the authors would find the study 'Generation of Sharp Wave-Ripple Events by Disinhibition' (Evangelista et al., J Neurosci 2020) interesting. This study contains a mechanism for the generation of SPW/R events by disinhibition of PCs by a third inhibitory cell population. Here, short-term depression acts on the connection between PVBCs and the third cell population.

We thank the Reviewer for drawing our attention to this interesting paper. A proper exploration of all possible termination mechanisms is a major task that we have postponed to a follow-up study. For now, we have removed the reference to our preliminary results, and added a citation to the article mentioned by the Reviewer in lines 510-516.

Table 3: The decay time constant for PC-PC synapses is large (more than 5 ms). Is this a realistic value for AMPAergic synapses?

Yes, this value is not just realistic, but it was determined experimentally for these connections by Guzman et al., Science, (2016).

(S) l. 604/ 620 (and also the section beginning on l. 291): The network was optimized to generate strong ripple-frequency oscillations in both the PC and PVBC population (Equation 9). In Figure 1D, however, it looks as if there was no ripple oscillation in the PC rate, but only in the LFP. Could the authors please clarify this? It would be nice to see a magnification of the spike rasterplot at the top of Figure 1D. I also wonder whether there could be a ripple oscillation present in the spiking activity of PCs shown in Figure 8C. It isn't present in the LFP as shown in Figure 8D, but since the network was optimized to display ripple oscillations in the PC activity (Equation 9) and since there still is bi-directional replay in the model, it might well be that there are still ripple oscillations in the PC population activity, as shown in Figure 3—figure supplement 1 for a different setup. It is also not defined how the population rates for PCs and PVBCs are calculated, but the temporal resolution (for example in Figure 1D middle plot) looks rather low and this might mask fast oscillations. Could the authors comment on this?

Ripples are present in the PC rate as shown in Supplementary Figure 5 (please also refer to our earlier responses). The population rate is calculated in 1 ms-long time windows as described in Methods line 724. On the other hand, plots showing the long-term population rate (such as the middle plot of Figure 1D) use a lower temporal resolution of 10 ms. The Reviewer is right that one cannot judge the presence or absence of ripple frequency oscillation from these plots, but we intended those panels to show the overall increase in PC population rate during sequence replay. On the other hand, Supplementary Figure 5 explicitly documents the presence of ripple oscillations in PC spiking activity.

Figure 2, supplement 2: it looks like some fraction of PC and PVBCs is bursting (very small ISIs). Are they important for the network dynamics? Is the adaptation maybe only effective in the bursting PCs? This might be important to gain a better mechanistic understanding of the model.

We have added panel C to Supplementary Figure 4, which shows that the low ISI values in PCs are due to some place cells firing more than once during a sharp wave (see the voltage trace of the second place cell). To further clarify this point, we have added an insert to panel B1 as well, showing that the lowest PC ISIs are above 5 ms (in fact, 5.96 ms is their refractory period (see Table 2)). Therefore, PCs in our model do not burst according to the usual definition (ISIs below 5 ms), but some of them do fire more than once during a sharp wave. As we argued above, we do not think that these multiple spikes are critical for the qualitative behavior of the network model. In principle, one could re-optimize the whole network with an extra feature controlling the number of PC spikes to judge how important this feature is, but we have not done so in the interest of time. However, we have added lines 186-191 to the results stating that this is a potential difference between the current model and the available data.

Reviewer #3 (Recommendations for the authors):The authors are careful to consider some of the limitations of their simple model. However, their study would certainly be strengthened if some of these and other questions could be addressed. Specific consideration should be given to the following:Spike adaptation alone does not result in a match to the experimentally-recorded durations of SWRs, as the authors acknowledge. It would be good at least to see results from the simulations that have been tried using an extra inhibitory neuron type to try to address this mismatch; and other mechanisms also could be explored here.

The mechanisms responsible for the termination of SWRs are currently unknown, and this is also an open question for our current model. Several different mechanisms based on cellular adaptation, short-term synaptic plasticity, and multiple inhibitory cell populations have been proposed. An additional possible mechanism based on the (learned or hard-wired) structure of recurrent excitation is demonstrated in this paper. As we briefly mentioned in the previous version of the manuscript, we also found that adding a second inhibitory population that provides delayed, long-lasting negative feedback to the PCs could explain the termination of SWRs. However, these simulations were performed in an earlier version of the circuit model that lacked learned excitatory connections, and we found that introducing this mechanism in the final model required substantial retuning of several parameters. Therefore, we decided to defer the full exploration of possible SWR termination mechanisms to a subsequent study; we have removed the reference in the manuscript to the preliminary simulation results, and cite some related published work (Evangelista et al., J. Neurosci. 2020) to back the proposed idea.

What specific spatio-temporal characteristics of spike activity trigger a sequence replay? For example, how sensitive is this to the number of closely associated place cells that spike contemporaneously and does it require some sequencing in this spiking?

This is an excellent question but, unfortunately, we do not know the answer yet. Simple observation of the spike rasters reveals no obvious pattern. One might guess that some combination of activity noise (stochastic activation of neurons) and “frozen noise” (randomness in connectivity and synaptic weights) determines where replay starts and the direction it takes (in the absence of external bias). However, it will require more sophisticated analysis methods or clever manipulations of inputs and weights to find a definite answer.

Is it such initial sequencing that determines whether forward or backward replay occurs?As discussed, there are biases towards forward or backward replay in different animal situations. How can such biases arise in this model? The "cued" replay results (Figure 2C) demonstrate the obvious of forward replay if the cue is at the track beginning, with backward replay if the cue is at the end of the track, but how in general may such "beginnings" and "ends" be defined in terms of place cells? The strict beginning and end here is somewhat artificial. How can a bias for forwards or backwards replay be introduced at intermediate points along the track? Insights into the characteristics of spike activity triggering replay, as asked for above, should help answer this.

In general, the direction of replay in the model can depend on systematic biases and random variations in the weight structure, but also on (random or deterministic) spatio-temporal structure in the external inputs. We believe that the cued replay scenario that we simulated is not entirely artificial – even real environments may contain choice points where the animal pauses to plan its route, or reward sites that again make the animal stop and may also activate subcortical modulatory systems that affect plasticity. At intermediate sites, systematic bias could be introduced during exploration by factors that influence behavior (e.g., running speed), neuronal activity or plasticity (e.g., attention), especially if such factors can change the fully symmetric nature of the synaptic plasticity rule. Random variations in connectivity or the weights (e.g., due to stochastic spiking during learning) will lead to a location-dependent systematic bias towards forward or reverse replay. Finally, (randomly occurring or input-driven) sequences in the external input are also expected to bias the direction of replay even at intermediate points. However, a detailed analysis of the relative impact of these factors will require further efforts. We have included some of these arguments in lines 404-427 of the Discussion.

Why do we not see simultaneous forward and backward replay from a single trigger point?

In fact, we do see some evidence of replay starting in both directions in the model in certain cases (please refer to the first and second detected replays in Figure 2A). However, there is competition between the two active populations (mediated by feedback inhibition), and one of the sequences always dies out after a short period (this is also because mutual support through recurrent excitation diminishes as the two active populations move further away from each other). This is similar to the competitive behavior of other “bump attractor” models. We have included these arguments in lines 159-164.